# Deep learning guided design of protease substrates

Carmen Martin-Alonso[1,2,10], Sarah Alamdari[3,10], Tahoura S. Samad[1], Kevin K. Yang [3], Sangeeta N. Bhatia [1,2,4,5,6,7,8,9,11] ✉ & Ava P. Amini [3,11] ✉

Proteases, enzymes that play critical roles in health and disease, exert their function through the cleavage of peptide bonds. Identifying substrates that are efficiently and selectively cleaved by target proteases is essential for studying protease activity and for harnessing it in protease-activated diagnostics and therapeutics. However, the vast design space of possible substrates (c.a. $20^{10}$ amino acid combinations for a 10-mer peptide) and the limited accessibility of high-throughput activity profiling tools hinder the speed and success of substrate design. We present CleaveNet, an end-to-end AI pipeline for the design of protease substrates. Applied to matrix metalloproteinases, CleaveNet enhances the scale, tunability, and efficiency of substrate design. CleaveNet generates peptide substrates that exhibit sound biophysical properties and capture not only well-established but also previously-uncharacterized cleavage motifs. To control substrate design, CleaveNet incorporates a conditioning tag that steers peptide generation towards desired cleavage profiles, enabling targeted design of efficient and selective substrates. CleaveNet-generated substrates were validated experimentally through a large-scale in vitro screen, even in the challenging case of designing highly selective substrates for MMP13. We envision that CleaveNet will accelerate our ability to study and capitalize on protease activity, paving the way for in silico design tools across enzyme classes.

Proteases are a diverse class of enzymes that play critical roles in both health and disease, including in coagulation, tissue remodeling, and cancer[1–3]. Because proteases exert their function through the cleavage of peptide bonds, studying and capitalizing on protease activity relies on the identification of peptide substrates that can be used as molecular probes[4,5], peptide-based inhibitors[6–8], or conditionally-activated triggers in engineered diagnostics[9–11] and therapeutics[12–15]. Yet, designing substrates that are both efficient—i.e., having high absolute cleavage—and selective—i.e., preferentially cleaved by a target protease over others—remains a significant challenge rooted in the complex biochemistry of proteases[16,17] and their substrates[18–20]. To achieve such diverse functions, proteases have evolved to display a broad range of cleavage specificities, mediated by interactions between a protease active site and a natural or synthetic substrate, often up to 10 amino acids long[21,22]. Substrate design is thus a combinatorial problem: for a 10-amino acid peptide, approximately $20^{10}$ (c.a. $10^{13}$) sequences

[1]Koch Institute for Integrative Cancer Research, Massachusetts Institute of Technology, Cambridge, MA, USA. [2]Harvard-MIT Division of Health Sciences and Technology, Massachusetts Institute of Technology, Cambridge, MA, USA. [3]Microsoft Research, Cambridge, MA, USA. [4]Institute for Medical Engineering and Science, Massachusetts Institute of Technology, Cambridge, MA, USA. [5]Marble Center for Cancer Nanomedicine, Massachusetts Institute of Technology, Cambridge, MA, USA. [6]Broad Institute of MIT and Harvard, Cambridge, MA, USA. [7]Department of Electrical Engineering and Computer Science, Massachusetts Institute of Technology, Cambridge, MA, USA. [8]Wyss Institute at Harvard University, Boston, MA, USA. [9]Howard Hughes Medical Institute, Cambridge, MA, USA. [10]These authors contributed equally: Carmen Martin-Alonso, Sarah Alamdari. [11]These authors jointly supervised this work: Sangeeta N. Bhatia, Ava P. Amini. ✉e-mail: sbhatia@mit.edu; ava.amini@microsoft.com

are possible, even without accounting for non-natural amino acids. Additionally, given that functionally related proteases often evolved from a common ancestor, they tend to share overlapping substrate sets, making it challenging to identify substrates that are truly selective for a target protease.

Given that all possible protease-substrate pairs have not yet been comprehensively profiled, substrate design typically involves two steps. The first is the nomination of candidate substrate sequences from a large, combinatorial space of possible sequences; the second is the selection of substrates as a function of their cleavage profiles after screening against proteases of interest. The most common approach for substrate nomination entails surveying the literature for existing substrates, informed by cleavage sites in naturally occurring proteins[23]. However, this search is inefficient and seldom results in synthetic substrates with desired cleavage profiles[19]. Alternatively, rational design leveraging expert knowledge of chemical biology can optimize synthetic substrates for specific targets, but this process is inherently resource-intensive, low-throughput, and bespoke to each protease of interest[24].

To circumvent bottlenecks in substrate nomination, recent technological advances have significantly increased the throughput of the selection step[25]. Large-scale chemical or display-based libraries, such as mRNA, ribosome, and phage display, have enabled high-throughput in vitro screening of $10^5$–$10^{13}$ unique substrates against proteases of interest[26-28]. However, these methods are sophisticated and costly, and thus not broadly usable. In contrast to high-throughput experimental approaches, computational methods offer the possibility of accelerating substrate selection through virtual screening. Methods leveraging inferred substitution matrices[29-31], structural and energetics patterns[32,33], or supervised learning[34-38] have focused on identifying substrate cleavage sites to predict whether a substrate will be recognized by a target protease. However, these methods return a binary cleavage prediction (i.e., cleaved vs. not cleaved), making it challenging to rank-order substrates by their cleavage efficiencies against a target protease, which is critical for engineering applications and the study of nuanced protease specificities. Recent deep learning models

that extract patterns from amino acid sequence data have enabled the accurate prediction of protein sequence-function relationships[39] and the generative design of proteins[40-42], but remain challenging to apply to enzymes and have been underexplored in their application to protease substrate design[43-45]. In silico methods that learn from large-scale screening data to design protease substrates with pre-defined cleavage preferences would increase the scale, diversity, and tunability of substrate design.

Here, we present CleaveNet, an AI-based pipeline for the end-to-end design of protease substrates (Fig. 1) CleaveNet consists of a predictive model, the CleaveNet Predictor, which assigns cleavage scores across proteases of interest (Fig. 1A), and a generative model, the CleaveNet Generator, which produces peptide sequences either unconditionally or conditioned on a desired protease cleavage profile input to the model (Fig. 1B). We validate our approach on the widely-studied family of matrix metalloproteinases (MMPs), which play important roles in health and a variety of diseases[3,46], by training CleaveNet on a dataset of mRNA-display peptides screened against 18 MMPs[47] and evaluating it both in silico and in vitro. We demonstrate that the CleaveNet Predictor achieves accurate and robust prediction of cleavage scores on an independent, experimentally-distinct test set and show that the CleaveNet Generator produces sequences that capture relevant cleavage profiles across MMPs and that exhibit both known and previously uncharacterized cleavage motifs. We use the end-to-end CleaveNet pipeline to design synthetic substrates for MMP13, a collagenase that plays a critical role in cancer metastasis, wound healing, and osteoarthritis[48,49], optimizing for both efficiency and selectivity (Fig. 1C), and experimentally assess their cleavage profiles through a large-scale in vitro screen of 95 substrates against 12 MMPs (Fig. 1D). All CleaveNet-designed substrates are cleavable by MMP13, and, strikingly, some substrates that were conditionally designed for high MMP13 selectivity are uniquely cleaved by MMP13. Our experimentally validated, AI-based design of protease substrates uncovers underappreciated determinants of MMP efficiency and selectivity, opening the door to overcoming the well-known promiscuity of MMPs. We envision that CleaveNet will advance the

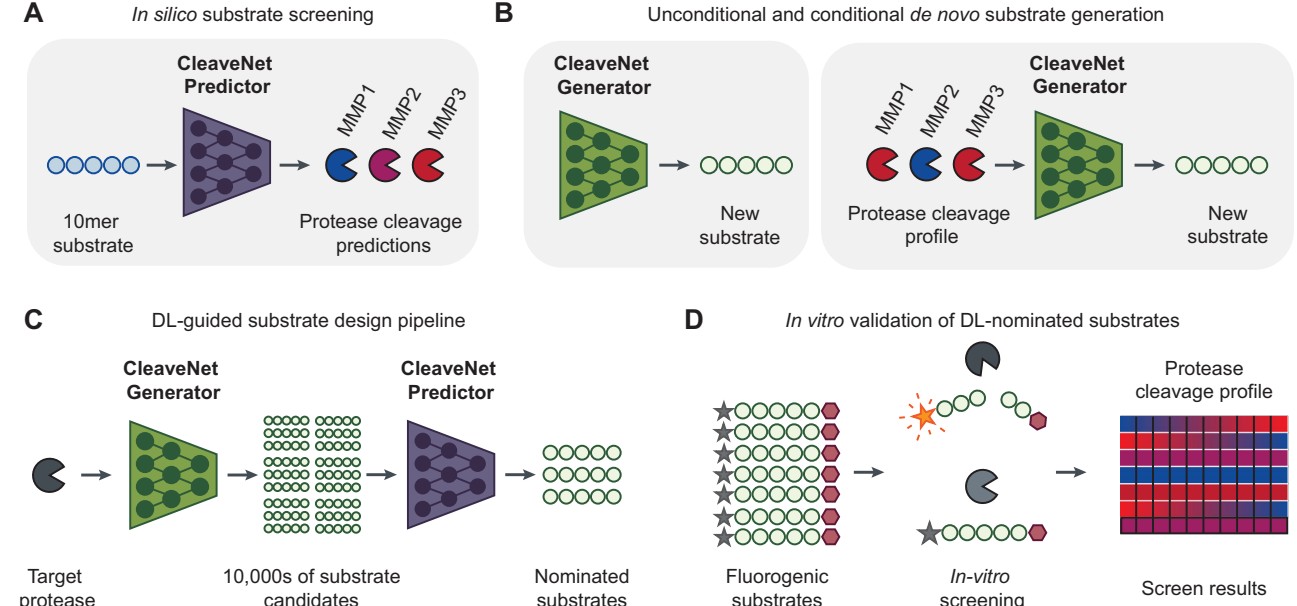

Fig. 1 | A deep learning approach for protease substrate design. A For a given peptide, the CleaveNet Predictor predicts cleavage scores across 18 matrix metalloproteinases (MMPs). B The CleaveNet Generator can generate substrates unconditionally (left) or conditionally, guided by a desired protease cleavage profile (right). C To design candidate substrates for a target protease, peptide sequences are generated by the CleaveNet Generator and prioritized by the cleavage scores from the CleaveNet Predictor. D An in vitro screen of 95 fluorogenic substrates against 12 recombinant MMPs was performed to validate the cleavage profiles of CleaveNet-designed substrates.

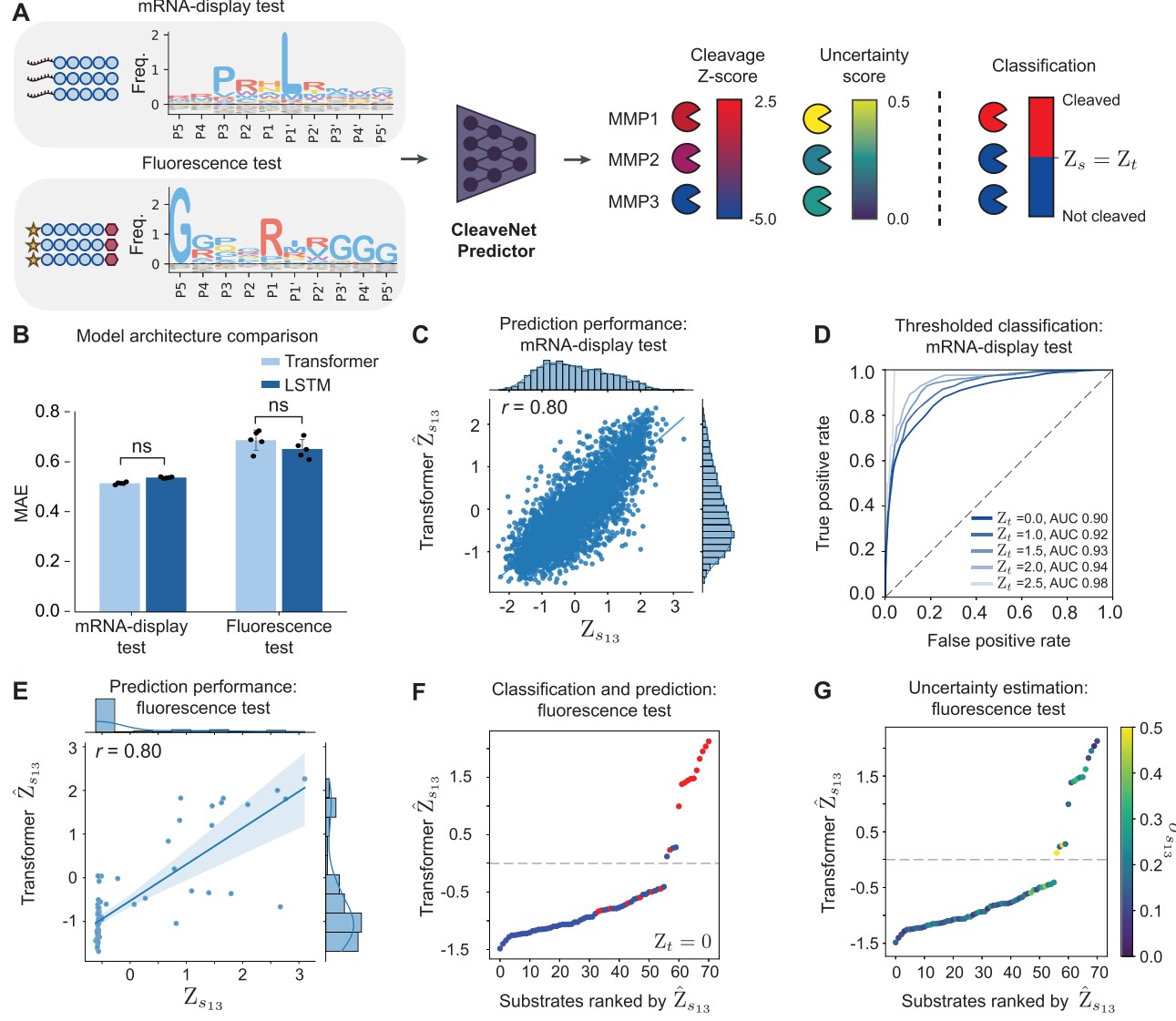

**Fig. 2 | CleaveNet accurately predicts cleavage efficiencies of synthetic peptides against MMPs. A** Multi-output cleavage score regression and uncertainty estimation by the CleaveNet Predictor. Two biochemically-distinct sets of synthetic peptides, from mRNA display ($n = 2901$ peptides) and fluorometric ($n = 71$ peptides) activity assays, are used as test sets for evaluation across different MMPs. IceLogos are normalized by natural amino acid frequencies (see "Methods"). Individual amino acids are colored by the chemical properties of their side chain: hydrophobic aromatic (lavender), hydrophobic (blue), hydrophilic (yellow), acidic (orange), and basic (red). **B** Mean absolute error (MAE) between true and predicted MMP-cleavage scores across the test sets, comparing the transformer (light blue) and LSTM (dark blue) models (mean ± s.d., $n = 5$ independent training trials; $^{ns}p > 0.05$,

one-way ANOVA). **C** Correlation of true ($x$-axis) and transformer-predicted ($y$-axis) MMP13-cleavage scores on the mRNA-display test set ($n = 2901$, Pearson's $r = 0.80$). **D** Receiver operating characteristic (ROC) evaluation of the CleaveNet transformer predictor over the mRNA-display test set for MMP13. Individual lines represent performance for different true cleavage score thresholds $Z_t$, with area under the curve (AUC) provided. **E** Correlation of true ($x$-axis) and transformer-predicted ($y$-axis) MMP13-cleavage scores on the fluorescence test set ($n = 71$, Pearson's $r = 0.80$). The data is a fit to a linear regression model; the resulting fit and a 95% confidence interval are also shown. **F, G** Fluorescence test substrates rank-ordered by predicted MMP13-cleavage score and colored by a binary indicator of true MMP13 cleavage (red, cleaved; blue, not cleaved; **F**) and by model uncertainty (**G**).

tunability, speed, and accuracy of protease substrate design across diverse applications.

## Results

### CleaveNet predicts cleavage efficiencies of peptides by MMPs

We trained our CleaveNet models on a publicly available dataset of c.a. 18,500 mRNA-displayed synthetic substrates and their continuous cleavage efficiencies across 18 MMPs, quantified as a normalized $Z$-score ($Z_{s_m}$) representing the relative strength of cleavage of substrate $s$ by protease $m$[47]. Although sequences with higher $Z$-scores can generally be expected to be cleaved with higher likelihoods, we note that this readout does not directly distinguish cleaved from non-cleaved substrates.

Since a critical component of substrate design is the down-selection of candidate substrates from a large combinatorial space, we first developed a predictive deep learning model, the CleaveNet Predictor, to score and virtually screen sequences in silico. We formulated the task of the CleaveNet Predictor as a multi-output sequence-to-function regression problem (Fig. 2A). Given an input amino acid sequence $s$, the model predicts continuous cleavage values $\widehat{Z}_{s_m}$ for 18 MMPs. Their associated uncertainty scores $\sigma_{s_m}$ are quantified as the standard deviation of predicted $Z$-scores measured by training an ensemble of five predictor models over the mRNA-display dataset. Continuous $\widehat{Z}_{s_m}$ values can optionally be converted to a binary classification output (i.e., cleaved vs. not cleaved) based on a desired threshold $Z_t$.

We investigated two model architectures common for sequence modeling—a recurrent bidirectional LSTM architecture[50] and a transformer architecture[51]. LSTMs learn patterns sequentially, while transformers learn patterns by looking at all elements of a sequence simultaneously and are the dominant architecture for protein language modeling[52,53]. To assess model performance and generalizability, we evaluated prediction performance across two datasets: sequences obtained from a 20% random split of the training dataset that were further filtered for homology against the training dataset (see "Methods") and never seen during training (mRNA-display test) and sequences obtained from an independent set of 71 Förster resonance energy transfer (FRET)-paired sequences that were previously screened against seven recombinant MMPs in vitro (fluorescence test)[54]. Substrates across the two test sets differed in length (10-mers vs. 7- to 14-mers for mRNA-display vs. fluorescence, respectively) and amino acid composition (Fig. 2A), and were assayed by very distinct experimental methods.

Both the LSTM and transformer models displayed similar predictive performance on the test sets, as supported by their comparable mean absolute errors between the true and predicted cleavage scores (Fig. 2B and Supplementary Table 1) and by the strong correspondence between predicted $\hat{Z}_{s_m}$ and true $Z_{s_m}$ for both models on the mRNA-display test set (Supplementary Figs. 1 and 2). With the goal of optimizing performance for substrates in the larger and more diverse mRNA-display dataset, we used the transformer-based model as the CleaveNet Predictor for subsequent analyses. The models performed better on some MMPs than others, with better performance achieved for MMPs that had either a higher fraction of training examples with high Z-scores or that displayed a wider dynamic range of Z-scores between cleaved and uncleaved sequences (Supplementary Fig. 3). These results suggest that our models may be more effective at learning cleavage patterns for proteases with more cleaved training examples or more specific cleavage patterns, respectively.

We observed strong correspondence between predicted $\hat{Z}_{s_m}$ and true $Z_{s_m}$ for MMP13 (Pearson's $r = 0.80$; Fig. 2C) and for other MMPs in the mRNA-display test set (Supplementary Fig. 2). To assess model performance for classification, we assigned true cleaved labels to sequences at varying Z-score thresholds: $Z_{s_m} \geq Z_t$, where {$Z_t = 0, 1.0, 1.5, 2.0, 2.5$}. We then calculated receiver-operator curves given a predicted $\hat{Z}_{s_m}$, over this range of possible cleavage thresholds. CleaveNet predictions were robust over the range of $Z_t$ evaluated, and classification performance improved as the Z-threshold increased, reaching an AUC value of 0.98 at $Z_t = 2.5$ (Fig. 2D and Supplementary Figs. 4, and 5); a threshold of 2.5 was previously reported to be broadly consistent with confident cleavage across all MMPs for this dataset[47].

To assess the robustness and generalizability of the models, we next evaluated their performance on the biochemically distinct fluorescence set (Supplementary Figs. 6–9). The Z-scores predicted by the transformer model displayed a strong positive linear correlation with true Z-scores (m = MMP13, Pearson's $r = 0.80$, Fig. 2E), especially for $Z_{s_{13}} > 0$ that are more likely to correspond to cleaved substrates. Given that $Z_{s_m}$ values are dependent on the nature of the experimental assay and cleavage range of the substrate library being profiled, which differ between the mRNA-display and fluorescence sets, the relative rank of the predicted Z-scores is a more appropriate metric to assess performance on the fluorescence test set. Independent of their exact value, we expect low Z-scores to be non-cleaved and higher scores to be cleaved. The CleaveNet Predictor consistently predicted lower $\hat{Z}_{s_m}$ for sequences that were not cleaved (blue) relative to substrates that were cleaved (red) (Fig. 2F). The uncertainty of the model coarsely correlated with the absolute error in $\hat{Z}_{s_m}$ (Supplementary Fig. 10) and reflected the confidence of individual predictions, with relatively low uncertainty for most non-cleaved substrates or top cleaved substrates (Fig. 2G and Supplementary Figs. 11 and 12).

Taken together, these results support the accuracy and robustness of the CleaveNet Predictor in producing MMP cleavage scores for synthetic substrates, even across biochemically distinct assay setups.

## CleaveNet designs biophysically plausible MMP substrates

Having validated the CleaveNet Predictor and with the aim of automating the substrate nomination step, we next sought to develop a generative model—the CleaveNet Generator—that would learn pan-MMP cleavage specificities and enable unconditional sampling of diverse MMP-cleavable substrates without further input. Ideally, generated sequences should capture the amino acid distribution and the biophysical and functional properties of training sequences, while still being diverse from each other and distinct from the training set. To generate sequences relevant to MMP activity, we trained an auto-regressive transformer model, which can generate variable-length sequences, on the training split of the mRNA-display dataset (see "Methods" for details). As a baseline, we generated sequences by sampling randomly and independently from the position-wise distribution of amino acids across all training sequences (referred to as the site-independent baseline, Fig. 3A).

We unconditionally generated 20,000 sequences using the CleaveNet Generator and the site-independent baseline and compared them to the mRNA-display test sequences (Fig. 3B, Supplementary Fig. 13). As expected for a method that considers each site independently, the baseline closely matched the test set position-wise amino acid distributions (average KL divergence = 0.237; Supplementary Fig. 13B). The CleaveNet Generator was effective at capturing the canonical MMP motif proline-X-X-hydrophobic residue and exceeded the site-independent baseline at recapitulating the amino acid distributions in positions P3 to P2′, which are most relevant for cleavage (position-wise KL divergence averaged over P3 to P2′ = 0.25 for CleaveNet-generated vs. 0.404 for site-independent baseline; Fig. 3B and Supplementary Fig. 13). The CleaveNet-generated and the site-independent baseline sequences exhibited biophysical properties consistent with those of the mRNA-display test dataset (Fig. 3C and Supplementary Fig. 14), supporting their quality and plausibility. Scoring each set of sequences with the CleaveNet Predictor (Fig. 3D) revealed that, across MMPs, the cleavage scores for the CleaveNet-generated sequences closely matched the distribution of scores for mRNA-display sequences, while the sequences from the site-independent baseline displayed significantly lower values across all MMPs (Fig. 3E).

To further evaluate the plausibility of generated sequences, we inspected the cumulative density function (CDF) of unique position-independent k-mers in each set of generated or mRNA-display sequences (Fig. 3F). Each of the mRNA-display, CleaveNet-generated, and site-independent baseline sequence sets effectively sampled the true 3-mer space, as demonstrated by saturation of the CDF by 8000 unique k-mers. However, as the space of possible unique k-mers became increasingly large for 4-, 5-, and 6-mers, the site-independent baseline covered the space near linearly, suggesting generation closer to random across the space of all possible k-mers. In contrast, the distribution of unique k-mers was more consistent between the mRNA-display and CleaveNet-generated sequences across all k-mer lengths, indicating that CleaveNet generations exhibit similar k-mer diversity to sequences from the mRNA display set. Inspection of the top occurring 5- and 6-mers in each set of sequences revealed that those from CleaveNet were closer to canonical MMP motifs than 5-mers from the site-independent baseline (Fig. 3F). Further, only 95 of the 20,000 sequences generated by CleaveNet were exact matches to train set sequences, indicating that CleaveNet was not simply memorizing training data. Together, these results demonstrate that the CleaveNet Generator can produce biophysically plausible sequences that exhibit expected cleavage profiles across MMPs and possess

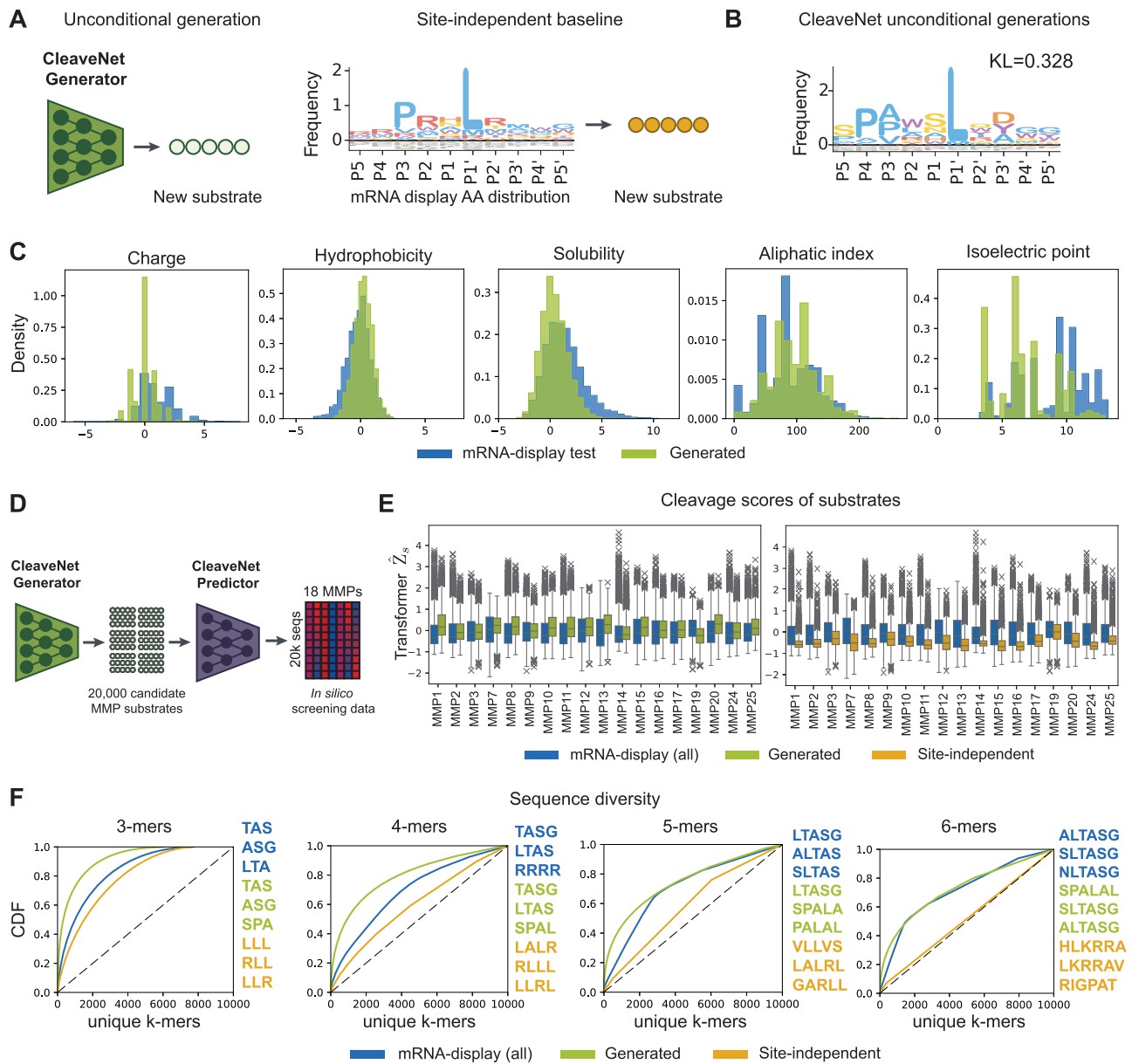

**Fig. 3 | CleaveNet generates biophysically plausible MMP substrates.**
**A** Unconditional generation from the CleaveNet Generator. Baseline sequences are produced via position-wise sampling from the amino acid distribution of peptides in the mRNA display train set. **B** IceLogo visualization of position-specific amino acid composition of CleaveNet unconditional generations ($n = 4000$), with the Kullback–Leibler (KL) divergence between the generated and mRNA-display test distributions denoted (normalized by natural amino acid frequencies). Position-wise distributions are included in Fig. 13. **C** Probability density functions of in silico-computed biophysical properties of generated (green, $n = 4000$) and mRNA-display test (blue, $n = 3717$) sequences. **D** CleaveNet was used to generate 20,000 candidate substrates and score their cleavage profiles across 18 MMPs.

**E** Distributions of predicted cleavage scores, across 18 MMPs, for CleaveNet unconditional generations (green, $n = 19,905$), site-independent baseline sequences (yellow, $n = 20,000$), and mRNA-display sequences (blue, $n = 18,583$). Box plots show the median (center line), interquartile range (box bounds, 25th to 75th percentile), whiskers extending to the most extreme points within 1.5 times interquartile range (or to the minimum or maximum, if none exceed), and outliers plotted as points beyond the whiskers. **F** Cumulative distribution of the frequency of unique $k$-mers present in CleaveNet generations (green, $n = 19,905$), site-independent baseline sequences (yellow, $n = 20,000$), and mRNA-display sequences (blue, $n = 18,583$) as a function of the total number of $k$-mers considered. The top-occurring $k$-mers in each set are provided (right).

diverse, distinct motifs while retaining sequence identities compatible with MMP cleavage.

## Generated sequences show biologically relevant cleavage patterns

To delve deeper into the biological validity of the CleaveNet pipeline, we virtually screened the 20,000 unconditional generations and analyzed the sequence determinants and activity profiles of top-scoring substrates across individual MMPs. First, we inspected position-wise determinants of specificity by plotting IceLogos, which reflect

position-specific amino acid compositions, for each set[55]. Despite containing no overlapping sequences, the amino acid profiles of top-scoring generated sequences closely matched those in the mRNA display test set for individual MMPs (Fig. 4A). Shared trends also emerged across MMPs within the same subclass (i.e., collagenases, gelatinases, MT1-MMP, MT2-MMP, and stromelysins). Although the PXXL motif was prominent across all MMPs, less expected amino acid preferences also emerged in the CleaveNet-generated sequences. Notably, a preference for methionine at P4 was identified (fold-increase above natural frequency of: 11.4-fold for gelatinases, 10.9-fold

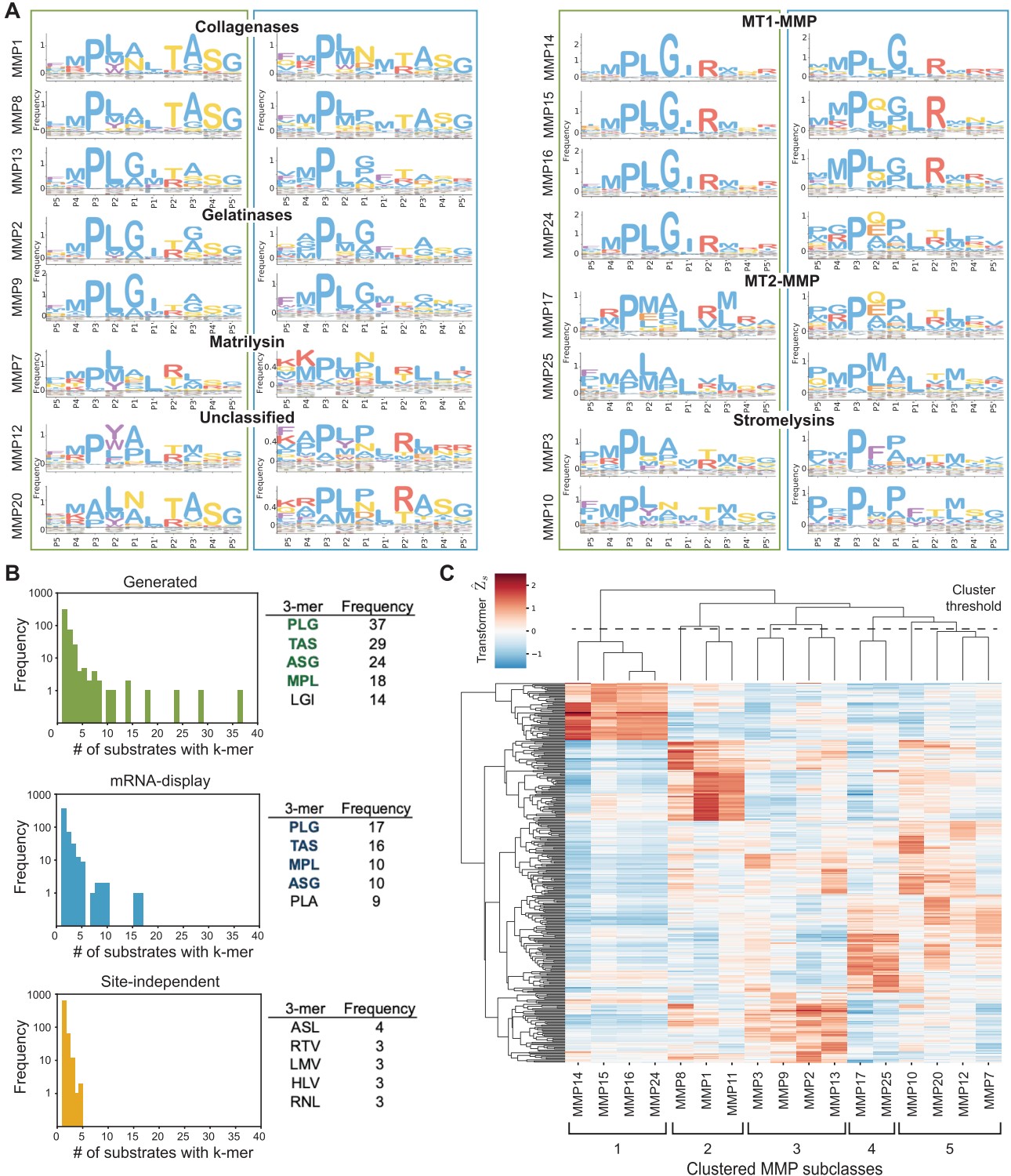

**Fig. 4 | Generated sequences recapitulate biologically relevant cleavage patterns across MMP functional classes. A** IceLogos for individual MMPs, grouped by relevant subclasses, are shown for the top 100 Z-scoring sequences in each set: CleaveNet-generated (green boxes) and mRNA-display training (blue boxes). The IceLogos are normalized by amino acid frequencies from the mRNA-display set. **B** Histograms of 3-mer frequencies shared across the top 100 Z-scoring MMP13 sequences in each of the CleaveNet-generated (top), mRNA-display training (middle), and site-independent baseline (bottom) sets. The frequencies of the top-five occurring 3-mers in each set are summarized as tables on the right, with shared 3-mers between the CleaveNet-generated and mRNA-display sets bolded. **C** Heat map, colored by CleaveNet-predicted cleavage score and annotated by hierarchical clustering, of the 25-top Z-scoring substrates per MMP, including the similarity cutoff and subsequent clustering of proteases into 5 groups with shared phylogeny.

for collagenases, 14.0-fold for MT1-MMP, 9.2-fold for MT2-MMP, and 12.2-fold for stromelysins), suggesting a previously unexplored relevance of this amino acid to the cleavage efficiency of short peptides for the MMP catalytic class.

Closer inspection of specificity determinants for the gelatinases MMP2 and MMP9, which have well-characterized cleavage preferences, revealed trends consistent to those previously reported[56,57] (Fig. 4A). For instance, we noted a higher prevalence of proline at P3

(99% of sequences) in the top Z-scoring generated sequences for gelatinases over other MMP classes, a preference for small amino acids such as glycine and alanine over aromatic and larger aliphatic residues at P1 and P3′, and a preference towards the positively charged amino acid arginine (27.5% of sequences) at P2′.

We next evaluated our pipeline's ability to learn complex inter-amino acid relationships critical to substrate recognition, such as subsite cooperativity. Such interactions between multiple subsites help enable cleavage but cannot be captured by position-wise IceLogos. To understand whether our pipeline was capturing such effects, we inspected the frequency and the identity of 3-mers shared across the 50 top-ranking MMP13 substrates in the CleaveNet-generated, mRNA display, and site-independent baseline sets (Fig. 4B). We observed significantly higher frequencies of some 3-mers over others for the generated and the mRNA display sets, suggesting a strong advantage for cleavage when specific amino acids are positioned contiguously. The canonical motif PLG emerged as the top occurring 3-mer for both datasets, and the top-4 occurring 3-mers were shared between the generated and mRNA-display sets, suggesting a high degree of meaningful conservation in generated sequences. In contrast, the site-independent baseline set displayed little preference towards any 3-mer and did not enrich for the canonical PLG motif. These results lend further support to the CleaveNet generation strategy and suggest that CleaveNet can learn complex inter-amino acid relationships that are otherwise overlooked by simply sampling from position-wise amino acid distributions independently.

As a final measure of biological validity, we used the CleaveNet Predictor to score the cleavage profiles of generated sequences, and then clustered the activity profiles of the top 25 Z-scoring sequences for each MMP (Fig. 4C). The cleavage profiles for all MMPs, except MMP11 and MMP12, clustered based on the phylogenetic distance of their catalytic domains (Supplementary Fig. 15)[58]. Group 1 consisted of the 4 transmembrane-type MT-MMP family members (MMP14, MMP15, MMP16, and MMP24); group 2 was collagenases (MMP1 and MMP8); group 3 included gelatinases (MMP2 and MMP9), the collagenase MMP13, and the stromelysin MMP3; group 4 was the glycosylphosphatidylinositol-anchored MT-MMPs (MMP17 and MMP25); and group 5 included the non-furin regulated MMPs MMP7, MMP10, and MMP20.

Altogether, these results reinforce the biological validity of CleaveNet sequences by demonstrating their alignment with well-established MMP cleavage preferences, their ability to capture subsite cooperativity patterns, and the clustering of their activity profiles according to the phylogenetic relationships of MMPs. Moreover, CleaveNet generated comparable—but non-overlapping—data to those obtained through the time and resource-intensive mRNA display screen used to collect the training data, providing a rapid in silico method to increase the scale and diversity of sequences that can be explored for tasks of interest.

### Generated substrates are cleaved by MMP13 in vitro
To further validate the CleaveNet pipeline, we used CleaveNet to nominate efficient substrates for MMP13 and tested them experimentally through an in vitro cleavage assay. To this end, we virtually screened the 20,000 unconditionally generated substrates with the CleaveNet Predictor and rank-ordered them by their uncertainty-aware predicted cleavage scores for MMP13 (Fig. 5A). To maximize the likelihood of identifying distinct substrates, we selected 24 substrates with the highest predicted MMP13 cleavage scores and non-overlapping 5-mers. As controls for this CleaveNet-guided substrate design pipeline, we included site-independent substrates that were obtained using the site-independent baseline alone or after rank-ordering with the CleaveNet Predictor. The latter group was included, given the poor performance of the site-independent baseline alone in previous analyses, and to assess the value of in silico screening with the CleaveNet

Predictor on substrates not designed by the CleaveNet Generator. The five top MMP13 substrates and five uncleaved substrates from the mRNA display training set were included as positive and negative controls, respectively. We synthesized these substrates as fluorogenic FRET-probes and screened them in vitro against recombinant MMP13 by measuring increases in fluorescence indicative of cleavage over time (Fig. 5A). Fold changes in fluorescence at end-point were calculated as a proxy for cleavage activity for each substrate.

For ease of interpretation, fluorescence fold changes were transformed to cleavage efficiencies, with a value of 0 for substrates that were not cleaved, a value of 1 for the substrate with the highest cleavage rate, $FC_{max}$, and a fractional cleavage efficiency between 0 and 1 for all other cleaved substrates. We first assessed the hit rate, defined as the fraction of substrates with a measurable cleavage in vitro, across the different sequence groups. In agreement with our CleaveNet-predicted cleavage scores, all CleaveNet-generated substrates were cleaved (100%, 24/24), whereas only one of the eight substrates nominated with the site-independent baseline was cleaved (12.5%, 1/8) (Fig. 5B). Complementing the site-independent baseline with the CleaveNet Predictor enriched for sequences with high predicted cleavage scores and increased the hit rate to 100% (8/8) (Fig. 5B).

We next assessed the absolute values of MMP13 cleavage efficiencies across the groups (Supplementary Table 3). In addition to significantly outperforming the site-independent baseline ($p < 0.01$), both CleaveNet-guided approaches yielded substrates with cleavage efficiencies superior to those of positive controls from the training set (median cleavage efficiencies of 0.22, 0.37, and 0.64 for the mRNA-displayed, CleaveNet-generated, and site-independent + CleaveNet Predictor groups, respectively; Fig. 5B and Supplementary Table 3). Across its two design groups, CleaveNet produced 18 substrates with MMP13 cleavage efficiencies higher than that of the most highly cleaved training substrate, DL57 (RMPLGLRAPA, efficiency 0.46).

To identify sequences that were cleaved preferentially by MMP13, we compared the IceLogos for all sequences cleaved by MMP13 ($n = 38$; 24 CleaveNet-generated, 1 site-independent baseline, 8 site-independent baseline + CleaveNet Predictor, and five training sequences) with the subset that displayed efficiencies greater than 0.6 ($n = 7$; 3 CleaveNet-generated, 4 site-independent baseline + CleaveNet Predictor) (Fig. 5C). Whereas MMP13 was able to cleave 38/50 substrates and thus tolerated a large diversity of sequences, those that were most efficiently cleaved displayed more constrained sequence preferences. The top seven best cleaved substrates shared PL at positions P3-P2, suggesting a strong bias of MMP13 for leucine at P2, which may have been previously overlooked[56]. The P1 and P1′ sites were restricted to hydrophobic amino acids, as is canonical for MMPs, but displayed greater diversity than P3 and P2, with P1 allowing glycine, alanine, or proline and P1′ allowing methionine, leucine, or isoleucine. Albeit less dominant, a preference for alanine at P4 and for alanine or aspartic acid at P3′ also emerged in the seven best-cleaved substrates. The 4-mers shared across the top seven sequences—namely APLG, LGLT, PLAM, PLGI, and PLGL—mostly overlapped with the P4-P2′ region, but the 4-mer TASG overlapped with the P2′-P5′ region. The top MMP13 substrate DL73 (LFPLAMMDMT) exhibited an unconventional MMP cleavage sequence, with phenylalanine at P4 and methionines at P1′ and P2′. The two next best cleaved sequences, DL6 and DL50, shared the 6-mer APLGLT. These observations of the CleaveNet-guided substrates indicate that positions beyond the canonical P3-P1′ region may still be important contributors to MMP13 efficiency. Performing a similar analysis with the training dataset revealed distinct and less conserved amino acid preferences at all positions, with the exception of proline at P3 (Supplementary Fig. 16). These findings suggest that CleaveNet-designed substrates reveal distinct sequence determinants of MMP13 efficiency beyond those that could be inferred from the training set.

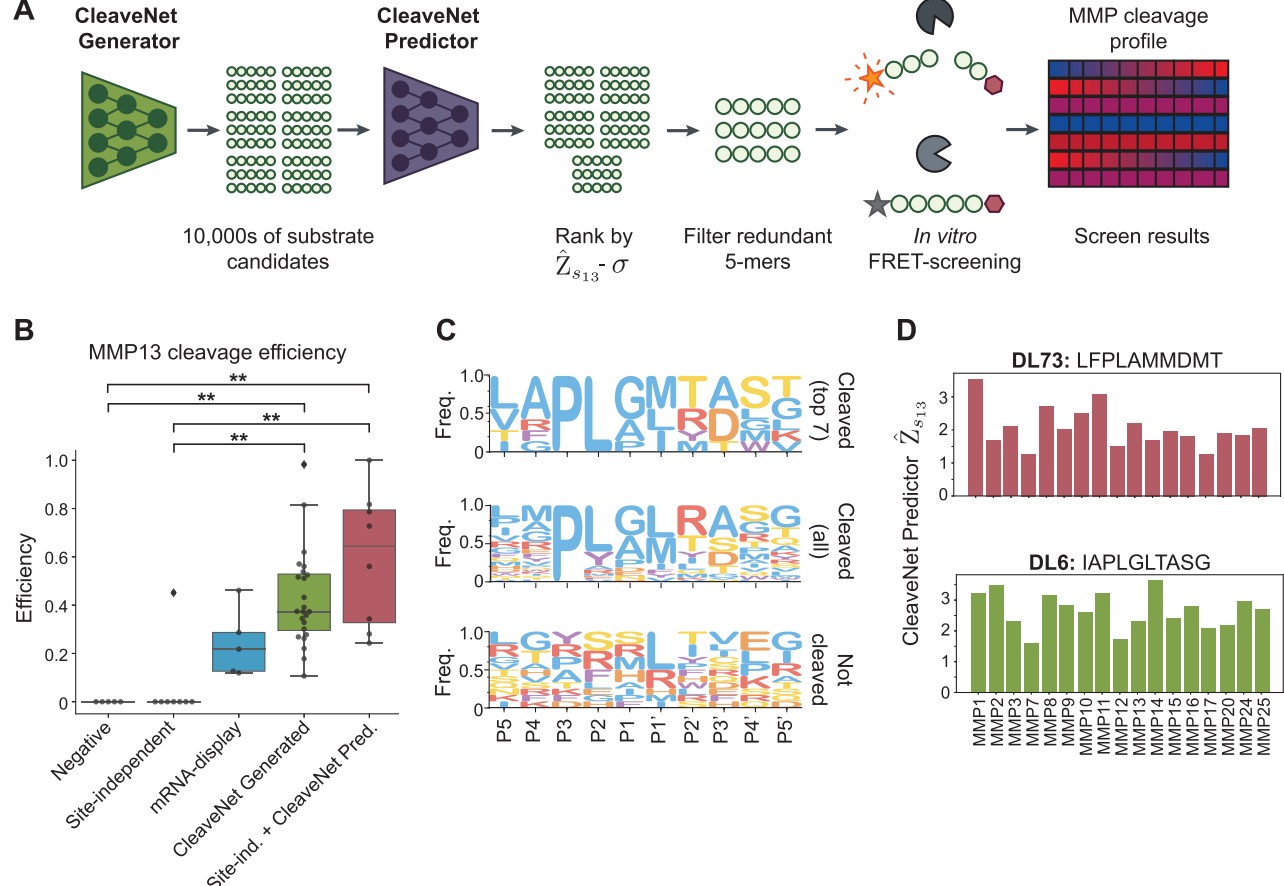

**Fig. 5 | CleaveNet-designed substrates are efficiently cleaved by MMP13 in vitro.** **A** Candidate sequences were generated by the CleaveNet Generator, ranked by uncertainty-aware MMP13-cleavage scores from the CleaveNet Predictor, and filtered based on redundant 5-mers. Nominated sequences were synthesized as FRET-substrates and screened in vitro against recombinant MMP13. Data were averaged across two independent runs, each containing duplicates of each reaction. **B** In vitro substrate cleavage efficiencies of CleaveNet-generated sequences as described in (**A**) ($n = 24$ sequences green), sequences from the mRNA-display dataset ($n = 5$ sequences each; negative controls, gray; positive controls, blue), and sequences from the site-independent baselines ($n = 8$ sequences each; site-independent alone, yellow; site-independent + CleaveNet Predictor, red). Box plots show the median (center line), interquartile range (box bounds, 25th to 75th percentile), whiskers extending to the most extreme points within 1.5 times interquartile range (or to the minimum or maximum, if none exceed), and outliers plotted as points beyond the whiskers. $^{**}p < 0.01$; Kruskal–Wallis test; exact values are provided in the Source Data file. **C** From top to bottom: IceLogo profiles, across all groups and displaying raw frequencies, of the top-seven substrates most cleaved by MMP13, all MMP13-cleavable substrates, and substrates not cleaved by MMP13. **D** Cleavage scores, from the CleaveNet Predictor, across all 18 MMPs for the two most efficient MMP13 substrates identified, DL73 (top; site-independent + CleaveNet Predictor group) and DL6 (bottom; CleaveNet-generated group).

To characterize the sequences of these seven highest MMP13 scoring substrates, we identified the sequences in training with which they shared the longest $k$-mers and their corresponding cleavage $Z$-scores (Supplementary Table 4). Sequences DL5, DL6, and DL50 shared 5-, 6-, and 8-mers with at least one training sequence with high $Z$-scores, suggesting a degree of overlap with training sequences. In contrast, DL73 (the top MMP13 substrate) and DL52 shared at most a single 4-mer with a small subset of training sequences exhibiting variable $Z$-scores. Further, the training sequences with the longest $k$-mer that matched DL3 and DL49 were not cleaved in the training set, supporting the distinction of some CleaveNet-guided sequences.

Performing in silico screening of the top two MMP13 sequences across other MMPs using the CleaveNet Predictor suggested that, albeit MMP13-cleavable, these substrates could also be susceptible to cleavage by other MMPs (Fig. 5D), making them inappropriate for applications where MMP13 selectivity is required.

Altogether, these results suggest that substrates designed for efficient MMP13 cleavage using CleaveNet-guided strategies are cleavable by recombinant MMP13 in vitro, overperform the site-independent baseline, and exhibit higher cleavage efficiencies and

distinct motifs than top-scoring substrates in the mRNA-display training set. However, these substrates are predicted to be cleaved promiscuously across multiple MMPs. To enrich for sequences with target cleavage profiles, such as high selectivity for MMP13, we next developed a guided generation approach.

**Conditional generation enables selective substrate design**
Identifying substrates that meet a predefined cleavage profile can be achieved by generating sequences unconditionally and filtering them based on their predicted cleavage profiles across MMPs. For instance, to design for substrates with high selectivity, sequences may be filtered by a selectivity metric calculated from predicted cleavage scores. However, this approach is untargeted (i.e., it may require a large number of unconditional generations to arrive at a selective sequence) and would be hard to generalize to nuanced cleavage patterns (e.g., if designing for substrates that are cleaved by multiple proteases of interest but not by others). To overcome these challenges, the CleaveNet Generator was trained with conditioning tags specifying target cleavage profiles across MMPs to guide generations to match desired profiles (Fig. 6A).

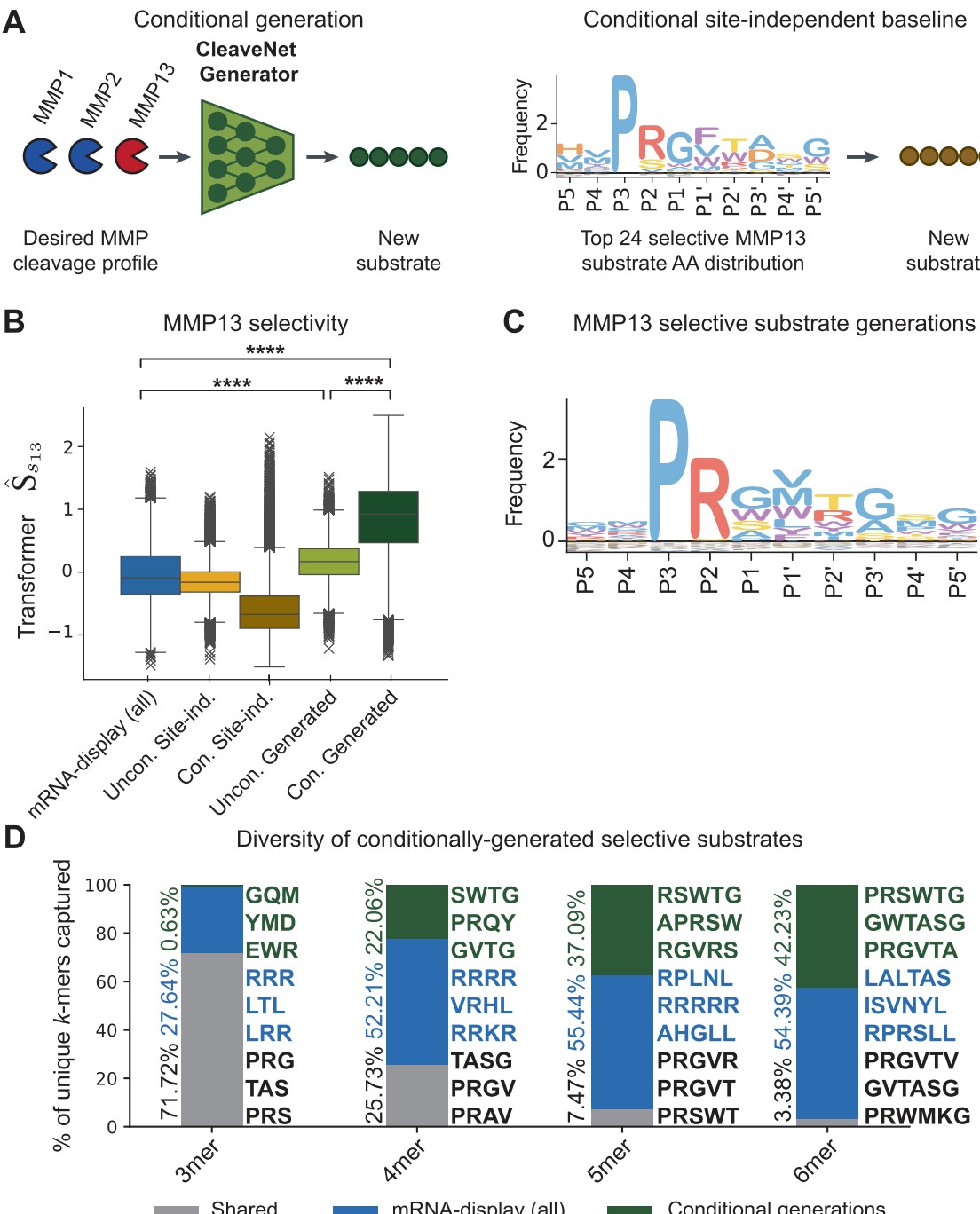

**Fig. 6 | Conditional generation enables targeted design of selective substrate sequences. A** The CleaveNet Generator can generate candidate substrates conditioned on a desired cleavage profile. Conditional generation from CleaveNet is compared to position-wise sampling from the amino acid distribution of the top 50 sequences that have the desired cleavage profile. **B** Predicted MMP13-selectivity scores, $\hat{S}_{s_{13}}$, for sequences conditionally generated by CleaveNet for MMP13 selectivity ($n = 20{,}000$, dark green), relative to sequences from the mRNA-display dataset ($n = 18{,}583$, blue), unconditional generations ($n = 19{,}905$, light green), and unconditional and conditional site-independent baselines ($n = 20{,}000$ each, light and dark yellow, respectively). Box plots show the median (center line), interquartile range (box bounds, 25th to 75th percentile), whiskers extending to the most extreme points within 1.5 times interquartile range (or to the minimum or

maximum, if none exceed), and outliers plotted as points beyond the whiskers. ****$p < 0.0001$; Kruskal–Wallis test. **C** IceLogo profile of the amino acid distributions for the CleaveNet-generated MMP13-selective sequences ($n = 20{,}000$ IceLogo normalized by natural amino acid frequencies). **D** Breakdown of shared and unique $k$-mers in a combined pool of mRNA-display sequences ($n = 18{,}583$, blue) and sequences conditionally generated by CleaveNet for MMP13 selectivity ($n = 20{,}000$, dark green). $k$-mers shared between the two sources are denoted in gray, $k$-mers only observed in the mRNA-display dataset are shown in blue, and $k$-mers only observed in the CleaveNet conditional generations are shown in dark green. The top three most frequently occurring $k$-mers in each subset are denoted to the right of each bar; the percentage covered by a given subset is denoted to the left.

To test this capability, we sought to use CleaveNet to design substrates selective for MMP13. We used the CleaveNet Generator to produce 20k sequences unconditionally or conditionally given a conditioning tag specifying high MMP13-selectivity. As baselines, 20k sequences were also sampled from the amino acid distribution of all training sequences (unconditional site-independent baseline) and from the distribution of the top 50 MMP13-selective sequences (conditional site-independent baseline). Here, the selectivity score refers to the cleavage of a sequence by one MMP relative to the average cleavage across all other MMPs, as previously described[47] (see "Methods").

Both sets of CleaveNet-guided sequences displayed significantly higher selectivity scores than the mRNA-display sequences ($p < 0.0001$; Fig. 6B). However, conditionally generated sequences achieved even greater selectivity than unconditionally generated ones ($p < 0.0001$; 5.5-fold higher median selectivity, Fig. 6B), highlighting the effectiveness of the conditional generation strategy. Accordingly, the substrates conditionally generated for MMP13-selectivity (Fig. 6C) were quite distinct from those in the mRNA display train dataset (Fig. 3A, IceLogo).

To investigate the distinctness of the conditionally generated sequences, we calculated the fraction of shared and unique $k$-mers in the combined pool of mRNA-display sequences and the sequences conditionally generated for MMP13 selectivity (Fig. 6D). While most 3-mers were shared, the fraction of shared $k$-mers decreased with $k$-mer length, leading to almost mutually exclusive sets of 6-mers between the two datasets. This observation indicates a divergence of CleaveNet's conditional generations from training sequences and raises the prospect of discovering motifs dictating MMP13 selectivity.

To demonstrate the generalizability of our approach, we used CleaveNet to design substrates selective for MMP9, a biologically distinct enzyme that plays direct functional roles in multiple hallmarks of cancer and has been a challenging target for the design of selective substrates, due to high overlap of substrate recognition with MMP2[57]. CleaveNet's conditionally-generated sequences achieved significantly greater MMP9-selectivity scores than both unconditional generations and samples from the site-independent baselines (Supplementary Fig. 17) and demonstrated substantial sequence divergence from the training set (Supplementary Table 5), supporting the generalizability of the CleaveNet pipeline to multiple MMPs. Together these in silico results suggest that the CleaveNet conditional generation strategy may offer a powerful tool to guide generations towards sequences with desired cleavage profiles.

### CleaveNet designs are selectively cleaved by MMP13

We next sought to validate experimentally that conditionally generated CleaveNet sequences indeed exhibited superior selectivity over other design strategies. To this end, we performed an in vitro screen of 95 FRET-paired, fluorogenic substrates ($n = 40$ selected for MMP13 efficiency, $n = 40$ designed for MMP13 selectivity, $n = 15$ total sequences from the mRNA-display set as controls for top efficient, top selective, and negative cleavage) against 12 recombinant proteases spanning all activity clusters in Fig. 4C (Fig. 7A and Supplementary Fig. 18). The 95-substrate panel was highly diverse in sequence space, as evidenced by a mean pairwise sequence similarity of 2.1 across the 95 tested substrates (Supplementary Figs. 19 and 20).

There was strong agreement between technical duplicates for each MMP ($0.74 < R^2 < 0.98$; Supplementary Fig. 21) and no dependence of cleavage on the properties of the crude peptides used (Supplementary Fig. 22). Moreover, given that cleavage events were detected for all MMPs in the fluorogenic screen, we could identify individual thresholds of predicted $Z$-scores associated with true cleavage for each MMP (ranging between CleaveNet-predicted threshold scores of 0.3 and 2.4; Supplementary Table 6 and Supplementary Fig. 23). Importantly, such cleavage thresholds could not previously be inferred from the mRNA-display dataset alone, as this dataset did not distinguish cleaved from non-cleaved substrates.

Visualizing the cleavage efficiencies of all protease-substrate pairs demonstrated that substrates designed for MMP13 efficiency were highly cleaved by MMP13 but also fairly promiscuously cleaved by other proteases, while substrates designed for MMP13 selectivity were more selectively cleaved by MMP13, consistent with design expectations (Fig. 7B and Supplementary Fig. 24). We next compared selectivity scores across the different MMP13-selective design groups (Fig. 7C and Supplementary Table 7). Similar to the results observed for MMP13 efficiency, both CleaveNet-guided approaches (conditional CleaveNet generated and conditional site-independent baseline + CleaveNet Predictor) outperformed the conditional site-independent alone baseline. However, given that some of the selective substrates from training were uniquely cleaved by MMP13 and thus had the highest selectivity score attainable, the CleaveNet-guided substrates at best only reached equivalent selectivity to these training examples.

Albeit highly specific, the MMP13-selective training substrates had relatively low efficiency scores ($E < 0.16$; Supplementary Table 7). We were thus curious to investigate how substrates from different groups compared as a function of both their cleavage efficiency and selectivity for MMP13. Plotting efficiency scores versus selectivity scores for each substrate enabled division of substrates into 4 quadrants[54]: those with low efficiency-low selectivity ($n = 56$), low efficiency-high selectivity ($n = 19$), high efficiency-low selectivity ($n = 15$), and high efficiency-high selectivity ($n = 5$) (Fig. 7D). There were three examples of generated substrates that achieved perfect selectivity for MMP13 (i.e., DL41, DL32, and DL28), matching the selectivity score of the best selective substrate in training (DL93) but at the expense of low efficiency (Fig. 7D, E).

A handful of substrates, like DL48, emerged in the upper right quadrant (Fig. 7D, E), characterized by much higher efficiency while maintaining high selectivity. By virtue of these properties, substrates in this quadrant may prove useful for engineering applications, such as the numerous conditionally-activated therapeutics in development[12–15]. Notably, this quadrant contained only fully CleaveNet-designed substrates. Additionally, all other substrates in this quadrant were distinct from the training set, with the closest train-set sequences seldom being cleaved by MMP13, but never selectively (Supplementary Table 8), with the exception of DL16, which shared a 4-mer with the top 3rd most selective substrate in training.

Closer inspection of the sequences in each quadrant of interest reinforced previously identified determinants of high MMP13 efficiency (Fig. 5C), while shedding light into determinants of MMP13 selectivity (Fig. 7F). Substrates with high selectivity but low efficiency were highly enriched for arginine at P2, aromatic residues, especially phenylalanine, at P1', and aspartic acid at P3', while those with both high efficiency and high selectivity shared traits from both.

Taken together, this large 95 substrate versus 12 MMP in vitro screen validated and calibrated our CleaveNet Predictor model across MMPs spanning all catalytic activity clusters, and successfully established the ability of the CleaveNet Generator to design substrates conditioned on a target cleavage profile. A subset of sequences conditionally designed for high MMP13 selectivity matched the maximum selectivity levels observed in training (low efficiency-high selectivity quadrant), while a different set exhibited higher cleavage efficiency while maintaining high selectivity (high efficiency-high selectivity quadrant), a cleavage profile that is desirable for engineering applications and that was largely absent from training.

### Discussion

Given that the vast repertoire of possible protease substrates is larger than what has been profiled to date, the identification of substrates with target cleavage profiles remains a significant challenge[8,18,19]. We developed CleaveNet, an AI-based pipeline for next-generation

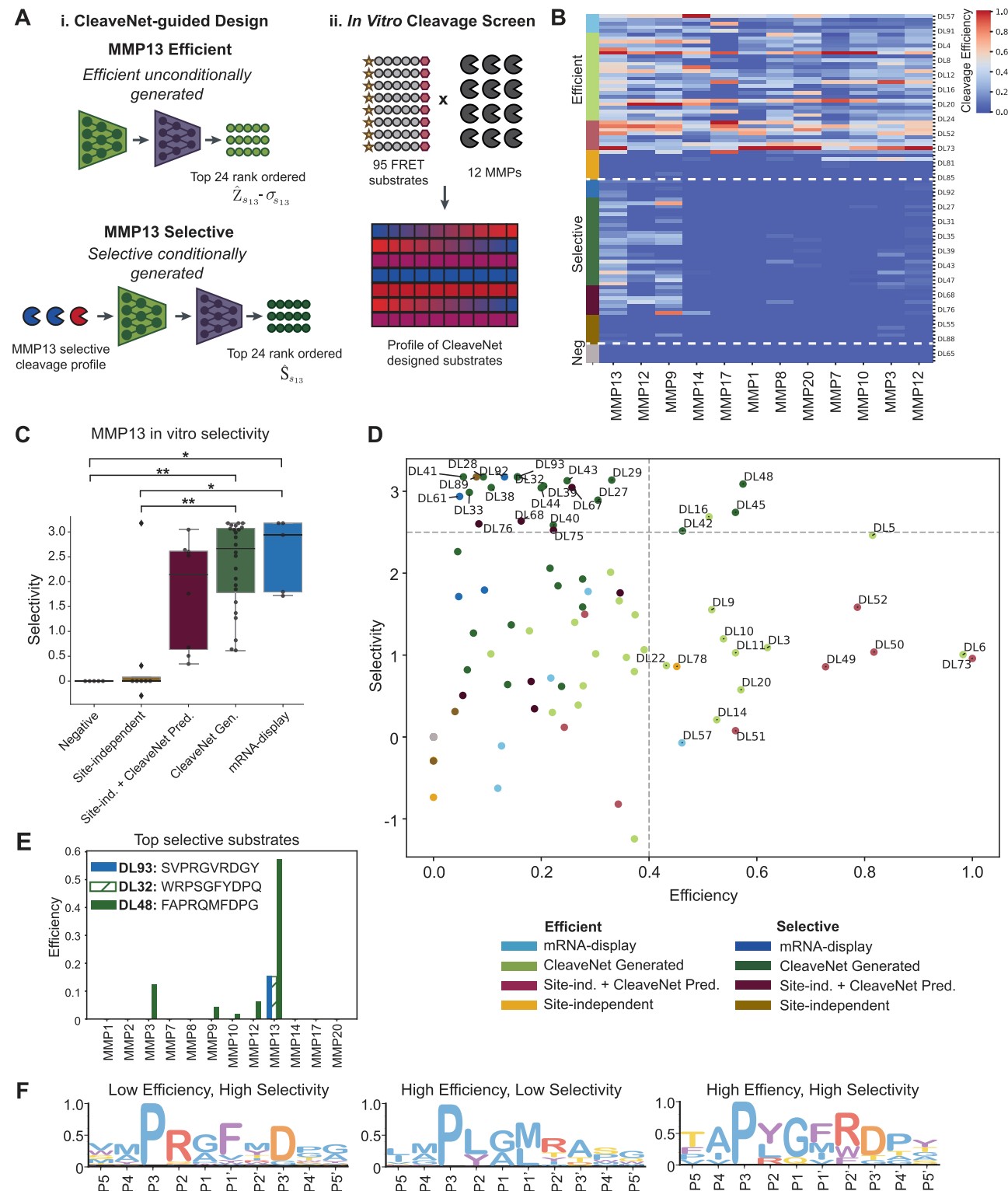

protease substrate design, that is capable of designing substrates with pre-defined cleavage profiles and that does not require prior expertise in chemical biology. The CleaveNet pipeline consists of a generator model and a predictor model that replace the traditional substrate nomination and selection steps and greatly increase the scale, tunability, and efficacy of the design process. We demonstrate the validity of our pipeline for the design of substrates for MMPs, a protease class critical in health and disease. Given that CleaveNet integrates continuous cleavage scores for a given substrate across 18 MMPs, it allows users to optimize for not only cleavage efficiency but also selectivity.

Using in silico analyses, we demonstrated that CleaveNet generates substrates that are distinct in sequence yet maintain comparable biophysical properties and cleavage profiles to training data. The pipeline was also able to recapitulate biologically relevant motifs across 18 MMPs and to capture intricate relationships dictating MMP cleavability, such as subsite cooperativity. Additionally, introducing a conditioning tag enabled guidance of substrate generation towards specific cleavage profiles, such as high MMP13 selectivity. We validated CleaveNet by experimentally testing substrates designed for high MMP13 efficiency or selectivity in vitro against 12 recombinant

**Fig. 7 | CleaveNet designs substrates that are selectively cleaved by MMP13 in vitro. A** Schematic overview of nomination strategies for substrates included in the in vitro screen (*n* = 95 substrates total). (i) Substrates were selected from unconditional generations for efficient cleavage by MMP13 (top) or conditionally designed to be cleaved selectively by MMP13 (bottom). In addition to CleaveNet-generated substrates (green, *n* = 24 per group), appropriate baselines consisting of site-independent baseline alone (yellow, *n* = 8 per group) and site-independent + CleaveNet Predictor (burgundy, *n* = 8 per group) were added; a schematic of these baselines is depicted in Fig. 18. Controls from the mRNA-display training set corresponding to substrates that were efficiently, selectively, or not cleaved by MMP13 (light blue, dark blue, and gray respectively, *n* = 5 per group) were also included. (ii) 95 FRET-paired substrates were screened against 12 recombinant MMPs in vitro. Data was averaged across two independent screens, each containing duplicates for each protease-substrate reaction pair. **B** Heatmap showing in vitro cleavage efficiencies for all protease-substrate pairs. Substrates were ordered by their expected MMP13 cleavage profile: efficient (top), selective (middle), uncleaved (bottom). **C** In vitro selectivity for substrates in groups designed for high MMP13 selectivity. Box plots show the median (center line), interquartile range (box bounds, 25th to 75th percentile), and whiskers extending to the minimum and maximum. *$p < 0.05$, **$p < 0.01$; Kruskal–Wallis test; exact values are provided in the Source Data file. **D** Efficiency versus selectivity plot for the 95 substrates, color-coded by group. Setting thresholds at $E > 0.4$ and $S > 2.4$ (dotted gray lines) divides substrates into four quadrants of activity. **E** Comparison of in vitro efficiencies observed for the single-most selective substrate from each of the mRNA-display group (blue) versus the CleaveNet conditionally-generated group, delineating substrates with high selectivity-low efficiency (green striped) versus high selectivity-high efficiency (green solid). **F** IceLogos characterizing substrates in distinct activity quadrants (IceLogos normalized by natural amino acid frequencies).

MMPs. This task, once exceedingly difficult due to the extensive substrate overlap of MMP13 with collagenases and gelatinases[47,56], remains essential for developing effective MMP13-specific profiling tools and MMP13-activated diagnostics and therapeutics. All CleaveNet-designed substrates were cleaved by MMP13. Among substrates designed for high MMP13 selectivity, we identified three substrates that were uniquely cleaved by MMP13 but exhibited low efficiency, comparable to the most selective substrates in training. Additionally, we discovered a distinct set of CleaveNet-designed substrates with considerable selectivity for MMP13 but significantly higher efficiency— a cleavage profile entirely absent in the training data. These findings highlight the potential utility of CleaveNet for oversampling sparse sequence spaces of interest to identify substrates with distinct motifs and desirable cleavage profiles. Our fluorogenic screen also enabled identification of cleavage thresholds that can be used to calibrate predicted *Z*-scores to true cleavage efficiencies for individual MMPs and to better compare relative cleavage across MMPs. Such thresholds, utilized by provided functions for correcting efficiency and selectivity scores, will greatly improve data interpretability, substrate selection, and model conditioning (Supplementary Fig. 25). We open source the CleaveNet models, datasets, and codebase to empower the broader research community.

The most significant contribution of our work is the design and deployment of generative models for protease substrate design, a process that to date has remained highly manual, dictated by trial-and-error, and plagued by low hit rates. Our AI-based approach enables capabilities beyond those of state-of-the-art display-based strategies for substrate design. By learning from peptide sequence data and incorporating a flexible conditioning tag, CleaveNet inherently embeds biological plausibility into library design and provides a method to steer generations towards sequences that exhibit a cleavage profile of interest across multiple proteases. Far from replacing display-based technologies, CleaveNet complements them and will serve as a hypothesis generator to guide library design and to maximize the information that can be extracted from sparse datasets, greatly decreasing the complexity and cost of screening experiments. Additionally, when used in isolation, the CleaveNet Predictor outputs continuous cleavage scores, as opposed to binary cleavage labels, and even outperforms state-of-the-art tools for binary substrate classification (Supplementary Table 9). Taken together, CleaveNet provides an end-to-end AI design pipeline for protease substrates, enabling in silico generation of synthetic datasets comparable in scale and quality to the highest throughput display strategies available.

Despite its successes, CleaveNet has several limitations that highlight avenues for continued work. First, CleaveNet is currently limited to synthetic substrates and could be extended to encompass natural substrates with appropriate training data or to design full-length proteins containing substrates for proteases of interest to expand beyond applications enabled by short peptide substrates. Such

work could benefit from integrating CleaveNet models with other AI-based protein structure prediction[59,60] and generative design[40,42] models. Second, CleaveNet is so far only validated for MMP substrate design. Efforts to create similar datasets and models across other protease subclasses could greatly enhance CleaveNet's reach and its utility. Finally, even with high predictive accuracy and generation quality, experimental validation remains critical to ensure the real-world applicability of designed substrates, and thus, CleaveNet does not completely replace the need for in vitro testing.

CleaveNet opens the door to numerous lines of future research. With respect to MMP13 substrate design, structure-based docking analyses could reveal greater mechanistic understanding of the molecular determinants of MMP13 substrate recognition, and the selectivity and efficiency of designed substrates could continue to be refined by such structural hypotheses, additional rounds of conditioning around top hits, or incorporation of non-natural amino acids. CleaveNet could be deployed to design MMP substrates for different use cases, such as the design or identification of substrates selective to other MMPs beyond MMP13. Moreover, CleaveNet is also compatible with use cases where multiple protease cleavage events are required, for instance, in the design of substrates that require cleavage by multiple MMPs co-expressed in a pathologic condition to maximize pro-drug activation[61,62] or for substrate logic design[63,64]. Given that CleaveNet is trained across 18 MMPs, it also offers a method for systematic dissection of the determinants, redundancies, and limits of MMP specificity, and is poised to uncover important insights in protease biology. Looking forward, combining high-throughput activity data and the CleaveNet modeling framework could enable the creation of a protease substrate atlas that characterizes the complete repertoire of substrates across protease classes beyond MMPs. Such an atlas would set the stage for similar datasets and models for substrate design across additional enzyme classes, such as nucleases, kinases, and phosphatases.

In sum, CleaveNet provides a streamlined approach for the design of peptide substrates that target specific proteases. We envision that the public availability of the open-source CleaveNet models and datasets will democratize substrate design, making it accessible to the multidisciplinary groups doing protease research that may otherwise not have access to sophisticated display-based strategies or deep-learning and chemical biology expertise. By exploring vast sequence spaces in silico, CleaveNet will catalyze protease research and the development of protease-targeted profiling, diagnostic, and therapeutic tools.

## Methods
### Datasets
A library of 18,583 10-mer peptide sequences profiled via mRNA display for their cleavage by 18 MMPs[47] was used for training and validation. Each protease-substrate pair is associated with a normalized

score ($Z_{s_m}$) representing the strength of cleavage of substrate $s$ by a protease $m$. An 80/20 split was performed on the data to create training and test sets, resulting in 3717 sequences in the test set. To assess overlap between the train and test datasets, the minimum Levenshtein distance between each test sequence and the training set was calculated. It was found that 816 test sequences closely matched the training set (distance < 3). These sequences were removed, leaving 2901 non-overlapping sequences as the homology-filtered test set. This set was held out from all models during training and is referred to as the mRNA-display test set. This test set was held out from all models during training and is referred to as the mRNA-display test set. An additional, independent out-of-distribution dataset containing 71 peptide substrates screened in vitro against recombinant MMPs[54] was used as an additional test set. This is referred to as the fluorescence test set.

### Training the CleaveNet predictors

Given an input sequence $s$, a neural network is tasked to learn a multi-task regression with the objective to predict a continuous value $\widehat{Z}_{s_m}$ for all 18 MMPs. The network takes a single sequence $s$ as an input, tokenized via indexing each of the canonical amino acids (plus an additional [PAD] token), indexed at 0. For the transformer models, a [CLS] token is added to the start of each sequence.

For the predictor task, two model architectures were evaluated: a bidirectional LSTM and a Transformer. A grid search was done over the following hyperparameters to nominate the best set of parameters for each model: batch size (32, 64, 128, and 256); model hidden dimension (16, 32, 64, and 128); model hidden layers (2, 4, and 6); and dropout rate (0, 0.1, 0.25, and 0.3). The set of hyperparameters that produced the lowest overall test loss was chosen for each model.

The final transformer model was a 56k parameter model comprised of a 2-layer encoder-only transformer (model dimension = 32) with 6 attention heads. The model was trained with positional encodings applied to inputs, using a batch size of 64. The output of the transformer was pooled by taking the representation of the [CLS] token before the final layer. The final LSTM model was a 44k parameter model comprised of a 2-layer fully-connected bidirectional LSTM (model dimension = 32), trained using a batch size of 32. Dropout was included after each LSTM layer ($d = 0.25$) to prevent rapid overfitting. Both models were trained with a 32-dimensional embedding layer.

The transformer predictor model was trained using the learning rate formula from the original transformer paper[51]:

$$ lrate = d_{model}^{-0.5} \cdot \min\left(step_{num}^{-0.5}, step_{num} \cdot warmup_{steps}^{-1.5}\right) \tag{1} $$

with a 4k step linear warmup, followed by a decrease proportional to the inverse square root of the step number; the LSTM predictor model was trained using a learning rate of 5e-3. Both models were trained using the Adam optimizer over 70 epochs on 4 NVIDIA A6000 GPUs. To quantify model uncertainty and performance, an ensemble[65] of five predictor models was trained over five independent 80/20 train/validation splits of the original training data set, and evaluated using the independent mRNA display test set. The uncertainty score is quantified as the standard deviation of the predicted $Z$-scores from each of these five models. The checkpoint with the lowest validation loss, per ensemble, was used for model evaluations. Both LSTM and transformer models performed well on the validation and test sets, and as such, both are included in the final set of CleaveNet predictor models.

### Training the CleaveNet generator

The generator model is an autoregressive model, which learns to predict the next amino acid residue $x_i$ in a sequence $s$, from previous residues $(x_1 \ldots x_{i-1})$. The probability of a sequence can be factorized as a set of conditional probabilities:

$$ p(x) = \Pi_{i=1}^{N} p(x_i | x_1 \ldots x_{i-1}) \tag{2} $$

The network takes in a single sequence $s$ tokenized via indexing of the canonical amino acids, plus a [START] and [STOP] token added to the beginning and end of the sequence, respectively.

Both a Transformer decoder and LSTM network were evaluated for this task. A grid search was done over the following hyperparameters to nominate the best set of parameters and model architecture for this task: batch size (32, 64, and 128); model hidden dimension (16, 32, 64, and 128); layers (2, 3); and dropout rate (0, 0.1, and 0.2). The set of hyperparameters that produced the lowest overall test loss for unconditional generation was chosen for model comparisons. We observed that the LSTM loss did not improve with an increased number of model parameters (test loss of 42k parameter LSTM: 2.283 vs. 25k parameter LSTM: 2.220, Supplementary Table 2). Alternatively, the transformer-based generator model's performance improved with scale and performed superior to a similar size LSTM network (test loss of 56k parameter transformer: 2.203 vs. 42k parameter LSTM: 2.283, Supplementary Table 2). As such, the decoder-only transformer was selected as the model architecture for the final CleaveNet generator.

The generator's learning capabilities were assessed on two tasks: unconditional and conditional generation. To train the model conditionally, samples were trained with a vector of 18 MMP scores, rounded to the nearest tenth, in place of the [START] token. The model performed much better on the conditional task, compared to the unconditional task (test loss conditional-only: 1.998 vs unconditional-only: 2.139, Supplementary Table 2). To maximize for a flexible and performant model, the final CleaveNet Generator was trained 50% of the time unconditionally and the other 50% conditionally on a set of 18 $Z$-scores rounded to the nearest tenth place. This 50–50 training scheme yielded a performance comparable to a conditional-only model by increasing model parameters of the joint unconditional-conditional model (test loss unconditional-conditional: 1.980 vs. conditional-only: 1.998, Supplementary Table 2). Accordingly, the final CleaveNet Generator model architecture has 3 decoder-only layers (328k parameters) with 64 hidden model dimensions, 6 attention heads, trained using a batch size of 128 and the transformer learning rate, Equation (1)[51], with a 4k step linear warmup. The model was trained on 4 NVIDIA A6000 GPUs for 50 epochs, at which the test loss began to plateau. The checkpoint with the lowest test loss was used for model evaluations. At inference, the trained CleaveNet Generator can generate sequences unconditionally, by prompting with a [START] token, or conditionally based on a $Z$-score profile, by prompting with a set of 18 $Z$-scores.

### Selectivity score calculation

To evaluate substrate selectivity towards an individual protease, selectivity scores were computed as previously described[47]. Briefly, selectivity scores were obtained by normalizing $Z$-scores across MMPs via the following equation:

$$ S_{s_m} = \widehat{Z}_{s_m} - \frac{\sum_{i=0}^{i=M, i \neq m} \widehat{Z}_{s_i}}{M - 1} \tag{3} $$

### Unconditional generations

To evaluate the CleaveNet Generator, 20k sequences were generated unconditionally by prompting the model with a [START] token. These were generated with a standard sampling temperature of 1 and a repeat penalty of 1.2. The repeat penalty was applied to penalize consecutive sampling of a single amino acid. New tokens were predicted given the previous tokens until a [STOP] token was reached. Because of this, generated sequences could be shorter than, equal to, or longer than 10

residues in length. Generated sequences shorter or longer than 10 residues in length were filtered out for comparison purposes; this filtered out 87 sequences. Given the smaller design space, sometimes the model would generate exact matches to the Kukreja dataset; 95 exact matches were generated and filtered out. This resulted in a total of 19,905 sequences used to measure generation metrics. For the purpose of comparing generation quality to the mRNA-display test set, this set of 19,905 generations was downsampled to 4000 sequences for a fair head-to-head comparison.

## Conditional generations

Conditionally-generated substrates were designed by prompting the CleaveNet Generator with a vector containing 18 $Z$-scores. To obtain seeds for conditional design, $Z$-score profiles were predicted for the entire mRNA display dataset using the CleaveNet Predictor. The top 50 $\widehat{Z}_{S_{13}}$ scoring substrates were selected as MMP13-efficient seed sequences, and the top 50 $\widehat{S}_{S_{13}}$ scoring substrates were selected as MMP13-selective seed sequences (100 total). Predicted $Z$-score profiles for these 100 seeds, rounded to the nearest tenth, were used to seed all conditional generations. Each $Z$-score profile was used to seed 400 generations each, with a sampling temperature of 1.2 and a repeat penalty of 1.2, resulting in a total of 20k efficient and 20k selective generations. To generate sequences for each of the efficient and selective baselines, sequences were produced via random, position-wise, independent sampling from the position-wise amino acid distribution of the top 50 MMP13-efficient and MMP13-selective designs, respectively.

## Selection of CleaveNet-generated substrates for in vitro experiments

A total of 48 CleaveNet-generated sequences were used for in vitro validation. Twenty-four of these were selected for predicted MMP13 efficiency from a pool of unconditionally generated sequences, and the other 24 were chosen for predicted MMP13 selectivity from a pool of conditionally generated sequences. Starting from a set of 20k unconditionally generated sequences, designs were ranked by an uncertainty-aware cleavage score defined as $\widehat{Z}_{S_m} - \sigma_{S_m}$ and filtered for diversity. To encourage a diverse final pool, the set of generations was reduced such that all 5-mers were unique by keeping only the top-scoring sequence corresponding to each 5-mer. The top 24 MMP13-efficient designs were selected after this procedure. For the conditionally-generated sequences, seeds for MMP13 selectivity were chosen by taking the top 50 MMP13-selective $Z$-score profiles from the mRNA-display dataset and used to prompt the generation of 20k designs, as previously described. Sequences were then ranked by the predicted selectivity score $S_{S_m}$, without any uncertainty filters, and then filtered for diversity as was done for the efficient designs. The top 24 MMP13 selective sequences were chosen. The sequences of the 48 CleaveNet-generated substrates, annotated by the generation procedure, are provided in Supplementary File 1.

## Selection of baseline and control substrates for in vitro experiments

Sequences sampled position-wise from the mRNA-display amino acid distribution were used as a baseline for the CleaveNet models (site-independent baselines). For the efficiency baseline, 20k sequences were sampled directly from the mRNA-display test distribution, then filtered for diversity as described in the "Methods" section "Selection of CleaveNet-generated substrates for in vitro experiments". From this pool, five sequences were randomly selected for in vitro testing; these are denoted as the efficient site-independent set. As a stronger baseline, site-independent baseline sequences were then evaluated with the CleaveNet Predictor and ranked by their uncertainty-aware cleavage scores. The top five highest-scoring sequences were nominated

as a second baseline, denoted the efficient site-independent + CleaveNet Predictor set. For the selective baseline, 20k sequences were sampled via site-independent, position-wise random sampling from the amino acid distribution of the 50 top MMP13 selective sequences in the mRNA-display dataset and filtered for diversity as previously described. From this pool, five sequences were randomly selected for in vitro testing; these are denoted as the selective site-independent set. As a stronger baseline, site-independent baseline sequences were evaluated with the CleaveNet Predictor and ranked by their selectivity score. The top five highest-scoring sequences were nominated as a second selective baseline, denoted the selective site-independent + CleaveNet Predictor baseline. For in vitro experiments, 15 control sequences—10 positive and 5 negative—were also tested. The positive controls were selected as the top five, each of selective and efficient MMP13-cleaved sequences from the original mRNA display dataset. The 5 negative controls were also chosen from the mRNA display dataset, these controls were reported to have negative $Z$-scores across all 18 MMPs.

## Biophysical property prediction

Biophysical properties (aliphatic index, hydrophobicity, Boman solubility, charge, and isoelectric point) were measured using the peptides.py package.

$K$-mer analysis was performed using the `paa.substrate.generate_kmers` and `paa.substrate.search_kmer` functions in the Protease Activity Analysis package[54].

## IceLogo representations

IceLogos were generated using the logomaker package. As appropriate for each use case and as specified in each panel, amino acid frequencies were either displayed as raw frequencies or after normalization by the frequencies of amino acids in nature or by the background frequencies of amino acids in the mRNA-display set. Amino acids were colored as a function of their properties into: (1) lavender for hydrophobic aromatic (F, W, and Y), (2) blue for other hydrophobic (A, I, L, M, P, V, and G), (3) yellow for hydrophilic (C, N, Q, S, and T), (4) orange for acidic (D, E, and H), and (5) red for basic (K and R).

## In vitro screening of peptides against recombinant MMPs

FRET-paired substrates (Mca-DNP) were synthesized by CPC Scientific as crude peptides. The full list of sequences, including N- and C-terminal modifications, is provided in Supplementary Data 1. Recombinant proteases were purchased from Enzo Life Sciences (human MMP1, human MMP2, human MMP3, human MMP8, human MMP9, human MMP10, human MMP11, human MMP12, human MMP14, human MMP20) or R&D Systems (human MMP17). For recombinant protease assays, fluorogenic substrates (10 µM final concentration) were incubated with recombinant proteases at 37 °C for 3–24 h, allowing for signal saturation for multiple peptides. Proteases were incubated at 10 nM, with the exception of MMP2 (75 nM), MMP10 (75 nM), and MMP7 (30 nM), due to their lower activity levels. A standard MMP buffer (50 mM TRIS, 10 mM CaCl2, 300 mM NaCl, 20 µM ZnCl2, 0.02% Brij 35, 0.1% BS at pH 7.5) was used for all MMPs, except for MMP3 that is active at a lower pH (50 mM MES, 10 mM CaCl2, 300 mM NaCl, 10 mM ZnCl2, 0.02% Brij 35, 0.1% BSA, pH 6). Activation was only required for MMP17 and was performed per the manufacturer's recommendations. Proteolytic cleavage of substrates was quantified by increases in fluorescence over time by a fluorimeter (Tecan Infinite M200 Pro) and analyzed with the Protease Activity Analysis package[54]. Each plate contained duplicate reactions for each protease-substrate pair, and two replicates corresponding to two independent runs with different peptide plates and protease preparations were performed. No data was excluded.

### Calculation of cleavage efficiencies from in vitro data

To facilitate the interpretation of cleavage rates across different MMPs, raw cleavage rates were transformed to cleavage efficiencies, with a value of 0 for substrates that were not cleaved, a value of 1 for the substrate with the highest cleavage rate $FC_{max}$ (fluorescent units/min), and a fractional cleavage efficiency between 0 and 1 defined as $\frac{FC_x}{(FC_{max}-FC_{min})}$ for all other cleaved substrates.

### Cleavage threshold selection

Cleavage thresholds reflect the $Z$-score values, predicted by the CleaveNet Predictor, under which the ROC-AUC for classifying cleaved versus non-cleaved was maximized, given the true in vitro cleavage data for the 95 substrates screened (Supplementary Table 6). Lack of commercial options to purchase 6/18 MMPs, namely MMP11, MMP15, MMP16, MMP19, MMP24, and MMP25, meant that the average cleavage threshold across the 12 MMPs screened (0.983) was imputed for these 6 MMPs.

### Calculation of corrected efficiencies

Predicted $Z$-scores can be transformed to corrected efficiency values using the `corrected_efficiency` function, provided in the `Corrected efficiencies and specificities` notebook, that leverages cleavage thresholds inferred from the in vitro screen. To achieve this, MMP-specific cleavage thresholds are first subtracted from $Z$-scores, such that adjusted values that are positive correspond with true cleavage, while negative values correspond to no cleavage. Next, adjusted values are transformed to corrected efficiency values according to the equation below. After said transformation, the sequence with the highest adjusted $Z$-score for a given MMP has a value of 1, a fractional efficiency between 0 and 1 is assigned to all other cleaved substrates, and a value of 0 is assigned to non-cleaved substrates. $s$, $m$, $Z_{s_m}$, and $T_m$ correspond to a substrate, an MMP, a $Z$-score, and a cleavage threshold, respectively.

$$\text{Corrected Efficiency}\left(\widehat{E}_{s_m}\right)=\begin{cases} \dfrac{\widehat{Z}_{s_m}-T_m}{\left(\widehat{Z}_{s_m}-T_m\right)_{max}}, & \widehat{Z}_{s_m}>T_m \text{ (Cleaved)} \\ 0, & \text{otherwise (Non}-\text{cleaved)} \end{cases}$$

$$(4)$$

### Calculation of corrected selectivity scores

Given that corrected efficiency scores bring all MMPs back to the same playing field with values between 0 and 1, relative cleavage differences between substrates can be more effectively compared. The `corrected_selectivity` function, provided in the `Corrected efficiencies and specificities` notebook, leverages corrected efficiency values to calculate corrected selectivity values according to the equation below. $s$, $m$, and $E_{s_m}$ correspond to a substrate, an MMP, and a corrected efficiency, and $M$ is the total number of MMPs over which the selectivity score is being computed.

$$\text{Corrected Selectivity}\,(CS_{s_m})=\widehat{E}_{s_m}-\frac{\sum_{i=0}^{i=M,\,i\neq m}\widehat{E}_{s_i}}{M-1} \qquad (5)$$

### Benchmarking cleavage classification to ProsperousPlus

Classification performance (identification of cleaved versus non-cleaved) was compared for the 95 substrates we tested in vitro across the 8 MMPs available on ProsperousPlus (MMP1, MMP2, MMP3, MMP7, MMP8, MMP9, and MMP14). To classify substrates using CleaveNet, substrates were assigned a cleaved label if their predicted $Z$-score from the CleaveNet Predictor (transformer) was greater than their corresponding cleavage threshold in Supplementary Table 6. ProsperousPlus was also used to assign a cleavage probability. Since ProsperousPlus does not specify cleavage cutoffs for individual MMPs,

the default cleavage cutoff of 0.5 was used for this analysis. Sensitivity, specificity, and accuracy for individual MMPs were computed.

### Reporting summary

Further information on research design is available in the Nature Portfolio Reporting Summary linked to this article.

## Data availability

The raw data for this study and the data generated in this study have been deposited in a GitHub repository and are freely available at https://github.com/microsoft/cleavenet. The source data for individual figures are provided in the Source Data file. Source data are provided with this paper.

## Code availability

Code, model weights, generated sequences, and computed metrics are available at https://github.com/microsoft/cleavenet[66].

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

## Acknowledgements

The authors thank The Center for the Development of Therapeutics (CDoT) at the Broad Institute for their assistance with automated peptide plating; Nicolo Fusi for feedback on the project; Philip Rosenfield for assistance in project management; Hannah Richardson for assistance with software release; Heather Fleming for valuable feedback on the manuscript; and Melodi Anahtar for insight on efficiency versus selectivity analyses and support with PAA. C.M.A. acknowledges support from a fellowship from La Caixa Foundation (ID: 100010434, code: LCF/BQ/AA19/11720039) and a graduate fellowship from the Ludwig Center at MIT's Koch Institute for Integrative Cancer Research. T.S.S. was supported by a postdoctoral fellowship from the Ludwig Center at MIT's Koch Institute. Additional support was received from the Koch Institute's Marble Center for Cancer Nanomedicine. S.N.B. is a Howard Hughes Medical Institute Investigator.

## Author contributions

Conceptualization: C.M.A., S.N.B., and A.P.A.; Methodology: C.M.A., S.A., T.S., K.K.Y., S.N.B., and A.P.A.; Software programming: C.M.A., S.A., and A.P.A.; Experimental design: C.M.A., S.A., T.S., K.K.Y., and A.P.A.; Investigation: C.M.A., S.A., T.S., and A.P.A.; Validation: C.M.A., S.A., T.S., and A.P.A.; Formal analysis: C.M.A., S.A., T.S., and A.P.A.; Resources provision: S.N.B and A.P.A.; Data curation: C.M.A., S.A., S.N.B., and A.P.A.; Visualization: C.M.A., S.A., and A.P.A.; Writing—Original draft: C.M.A., S.A., and A.P.A.; Writing—Review & editing: C.M.A., S.A., T.S., K.K.Y., S.N.B., and A.P.A.; Supervision: S.N.B. and A.P.A.

## Competing interests

C.M.A. is an employee of Amplifyer Bio. S.N.B. reports compensation for consulting or board membership by Amplifyer Bio, Catalio Capital, Danaher, Earli Inc., Impilo Therapeutics, Matrisome Bio, Ochre Bio, Port Therapeutics, Ropirio Therapeutics, Satellite Bio, Sunbird Bio, Vertex Pharmaceuticals, and Xilio Therapeutics. All other authors declare no competing interests.
