## [Transparent Peer Review file · Nature Communications]

Deep learning guided design of protease substrates

Corresponding Author: Dr Ava Amini

Version 0:

Reviewer comments:

Reviewer #1

(Remarks to the Author)

This is an excellent, technically rigorous study that will be valuable to groups designing protease-responsive probes. The authors present CleaveNet, a two-part end-to-end AI pipeline tailored specifically for designing 10-mer peptide substrates targeting matrix metalloproteinases (MMPs). The CleaveNet Generator creates novel substrates, optionally conditioned on desired protease-cleavage profiles, while the CleaveNet Predictor evaluates cleavage efficiency and selectivity in silico. Both models were trained on extensive mRNA-display screening data from Kukreja et al. (2015), enabling careful uncertainty estimation via ensemble predictions. The comprehensive validation (including an extensive and informative 95-substrate in vitro screen focused on MMP-13) is particularly strong. Below, we highlight points that, if clarified or addressed through small additional analyses, could further bolster the manuscript's already substantial contributions.

Major comments

Cross-protease selectivity distortion from Z-score normalization

The manuscript relies heavily on Z-scores to quantify cleavage efficiency, compute selectivity, and condition the generative model. However, because Z-scores were computed independently for each protease (i.e., normalized using each protease's own mean and standard deviation) differences in underlying variance (σ) across proteases may introduce artifacts when comparing Z-scores across MMPs. This is particularly relevant for CleaveNet's selectivity metric, which depends on differences in Z-scores across proteases.

To illustrate clearly, consider a simple numeric example involving two hypothetical proteases (A and D):

- Protease A cleaves substrates x, y, and z with raw cleavage values of 70, 13, and 17, respectively (mean = 33.3, σ = 31.8).
- Protease D cleaves the same substrates with raw cleavage values of 98, 1, and 1 (mean = 33.3, σ = 56.0).

Substrate x receives an identical Z-score (1.15) from both Protease A and Protease D. Consequently, if selectivity scores are computed by subtracting the average Z-score of all other proteases (as defined in the manuscript), substrate x would yield very similar selectivity scores for both proteases, despite the stark differences in absolute cleavage levels and true biological selectivity. Protease D cleaves x far more selectively in absolute terms (raw = 98 vs. 1 and 1 for others), while A cleaves x more moderately and has a significant cleavage signal for the two other substrates (raw = 70 vs. 13 and 17 for others).

Because σ is larger for D (56.0) than for A (31.8), this real difference in selectivity is flattened. Conversely, the same mechanism could also inflate apparent selectivity if σ is small.

To be clear, this limitation affects cross-protease comparisons only. Within a single protease, Z-scores remain perfectly valid for ranking or classification. But for metrics like selectivity—which are computed from Z-score differences across proteases—variance heterogeneity may introduce distortions, potentially inherited by both CleaveNet's conditioning and evaluation procedures.

We recognize that the authors used Z-scores directly from Kukreja et al. (Chem. Biol. 2015), who applied a residual-based LOESS normalization to sequencing count data. While raw cleavage values were not made public, we believe it is reasonable to ask whether they can be requested from the original authors. We encourage the authors to:

- Reflect on how inter-protease differences in variance of raw cleavage data (σ) may bias selectivity scores.
- If feasible, attempt to obtain the raw data from Kukreja et al. to evaluate this directly. Alternatively, consider mitigation strategies such as estimating pseudo-raw cleavage magnitudes ($Z \times \sigma$) or adopting rank-based selectivity metrics that are more robust to scaling artifacts.

At minimum, a short discussion of these limitations would help readers interpret CleaveNet's selectivity outputs more cautiously and transparently.

Generalisation beyond MMP-13 and Benchmarking:

CleaveNet is framed as a family-wide design platform, yet all experimental validation targets MMP-13. Even a small in-silico case study on a biologically distinct enzyme (e.g. MMP-9 or MMP-14)—generating 10–20 candidate substrates, scoring

them with the Predictor, and showing predicted selectivity—would make the generalisation claim more credible. Specifically, the authors could demonstrate that CleaveNet reliably shifts predicted substrate performance above random peptide baselines and confirm sequence novelty relative to training peptides.

Additionally, a brief benchmarking comparison (if feasible) against established protease predictors (e.g., DeepCleave, PROSPEROUS, other) would help contextualize CleaveNet's predictive accuracy within the current state-of-the-art. If direct benchmarking is not feasible, clarifying why this is the case would strengthen the manuscript by clearly delineating CleaveNet's scope and comparative advantages.

Train–test sequence similarity.

The paper says the 18 583-peptide dataset was split 80/20 at random (Methods, p 40), so peptides differing by only a few residues may appear in both splits. Because short 10-mers can share identical stretches, this can over-estimate predictive performance. We recommend re-running the split with a simple clustering filter (e.g., CD-HIT at 70 % identity or removing any train–test pair with Levenshtein distance < 3) and reporting the resulting MAE/ROC. Even if the overall ranking is unchanged, showing robustness to a similarity-controlled split would strengthen the generalisation claim. (The external 71-peptide fluorescence set is a helpful check, but its limited size and coverage make the within-dataset control important as well.)

Uncertainty estimation and ensemble details.

The manuscript mentions that the CleaveNet Predictor is an ensemble of five models and that substrate selection is guided by an "uncertainty-aware cleavage score" ($Z_{sm} - \sigma_{sm}$). However, it remains unclear how the uncertainty score σ_{sm} is computed (e.g., whether it is the standard deviation of the ensemble predictions), and whether it correlates meaningfully with true prediction error. We suggest clarifying the method used to derive σ_{sm} , briefly describing the differences among the five models (e.g., random seeds vs. data splits), and if possible adding a calibration figure or summary statistic showing how σ_{sm} relates to observed error on the test set. This would strengthen the rationale for using uncertainty in substrate triage and enhance reproducibility.

Minor comments

- The text briefly notes that MMP-13 is implicated in cancer, wound healing and osteoarthritis, but doesn't say why its selectivity is unusually hard to achieve or therapeutically valuable. A short expansion in the Introduction or Results, e.g., noting its broad substrate overlap with other collagenases and current interest in disease-specific diagnostics, would ground the choice of showcase target.
- Readers first encounter the 10-mer length midway through the Introduction. Consider moving that statement up (e.g., into the Abstract or Figure 1 caption) so it's clear from the outset that CleaveNet currently designs fixed-length 10-residue peptides, even though 8-mer substrates are common in the protease literature. State explicitly whether k-mer counts ignore position. If positional context is not tracked, add a one-line note so readers don't assume site-specific motifs are being compared.
- The main text discusses DL42 as a top hit, whereas Table S4 lists DL52. Please identify which ID is correct and adjust the other location (or note that both are distinct examples).
- Table S1 and ROC curves cover all 18 MMPs, but only the LSTM has per-MMP scatter plots (Fig S2). Providing the same scatter panels for the transformer, or merging the two into a single comparative figure, would give readers symmetrical insight.
- Align Figure 7A caption with what is shown. The caption references yellow and burgundy baseline groups that are visible only in Fig S15. Either embed Fig S15 into the main text or trim the colour references in Figure 7A's caption so it accurately reflects the standalone figure.
- Substantiate "diverse" claim for screened peptides. A simple pair-wise similarity heat-map (e.g., Levenshtein distance or clustering dendrogram) for the 95 tested substrates would back up the statement that the experimental set is sequence-diverse.
- Fully label the heat-map in Figure 7B (or add an SI version). Only 24 of 95 substrates are currently annotated, which makes it hard to trace individual examples (DL41, DL48, etc.). A fully labelled version—either in the main figure or in Supplementary Information—would greatly improve interpretability.

Adrian Jinich, Sarah Veskimagi (UCSD)

(Remarks on code availability)

very solid repository. The repository already meets reproducibility guidelines and should be a usable resource for other groups.

small suggestion (sorry if it's already there and we missed it): Provide a small CSV of the 95 in-vitro peptides + raw cleavage values so users can reproduce Figure 7 directly.

Reviewer #2

(Remarks to the Author)

Martin-Alonso et al. in publication „Deep learning guided design of protease substrates” submitted to Nature Communications present an interesting in silico AI based approach to analyzing and predicting protease substrate specificity, focusing on modeling substrate–enzyme interactions. The methodology applied and the attempt to formulate a general predictive model are commendable, particularly in light of the growing importance of bioinformatics and AI in molecular biology.

However, the central research hypothesis—that substrate specificity can be reliably predicted using the proposed model—has not been convincingly validated. A critical limitation of the study is the exclusive use of a single group of proteases, matrix metalloproteinases (MMPs), which are known for their broad substrate specificity. Such enzymes are not suitable models for evaluating general mechanisms of substrate selectivity, as their activity does not follow well-defined or highly selective substrate recognition rules. Importantly, no MMPs exhibit the high substrate specificity required to serve as a robust basis for predictive modeling.

Furthermore, the *in silico* results have not been sufficiently or reliably compared and validated against existing experimental data, particularly those obtained using proteomic techniques and tools from chemical biology. The absence of such comprehensive validation significantly reduces the biological and predictive value of the proposed model. To substantiate the proposed approach, it is essential to test the model across a diverse set of protease classes, each representing different substrate-binding modes, hydrolysis mechanisms, and patterns of subsite specificity (for example, cathepsins, caspases, kallikreins and SENPs). A thorough analysis of subsite selectivity across all binding pockets is also necessary for each of these enzymes classes. Only by comparing *in silico* predictions with experimentally established specificity profiles—derived from proteomics and combinatorial chemistry—can the practical utility of the model be properly assessed.

In my opinion, the manuscript in its current form does not meet the publication standards of Nature Communications, as it does not confirm the research hypothesis in a comprehensive manner, nor does it provide sufficient validation of the findings. The authors have limited their analyses to a narrow scope, without contextualizing their results within the broader enzymology of proteases or making adequate use of available comparative data.

In conclusion, while the study may offer a thought-provoking theoretical concept and a potential starting point for future investigations, its current scope and level of validation make it more appropriate for publication in a specialized journal rather than as a full-fledged research article of broad relevance. Therefore, I do not recommend this manuscript for publication in its current form.

(Remarks on code availability)

Reviewer #3

(Remarks to the Author)

The authors report the development of sequence-based AI models for predicting and designing protease specificity, focusing on MMPs. The studies are designed creatively, performed carefully, interpreted thoughtfully, and reported succinctly.

For *in silico* evaluation of protein-related problems, there is always the question of data leakage leading to memorization effects. The experimental results showing efficacy mitigate this concern to a great extent, but instead of using Logo plots, the authors could perform additional analysis of how much training and test sets overlap and implications of that, as well as comment on dataset balancing?

It is great to see that the generated sequences have higher catalytic efficiency. Can the authors identify why this is the case? What is the expectation from chance?

Have the authors considered using structural models to gain insights into successful and failed predictions?

Also please cite the following protease specificity measurement and prediction works:

<https://pubmed.ncbi.nlm.nih.gov/23589865/>
<https://pubmed.ncbi.nlm.nih.gov/27932294/>
<https://pubmed.ncbi.nlm.nih.gov/30587591/>
<https://pubmed.ncbi.nlm.nih.gov/37729196/>
<https://pubs.acs.org/doi/10.1021/acssynbio.0c00452>

(Remarks on code availability)

Reviewer #4

(Remarks to the Author)

(Remarks on code availability)

Version 1:

Reviewer comments:

Reviewer #1

(Remarks to the Author)

We carefully reviewed the authors' responses and the revised manuscript. All our previous concerns have been fully addressed, and the revision is clear, rigorous, and complete. We have no further comments and recommend acceptance.

(Remarks on code availability)

Yes. The code is clearly documented, functional, and matches the analyses described in the paper. It provides sufficient instructions for reproduction and will be a useful resource for the community.

Reviewer #3

(Remarks to the Author)

The authors have satisfactorily addressed my comments.

(Remarks on code availability)

Fine.

Reviewer #4

(Remarks to the Author)

(Remarks on code availability)

Overview of changes

We thank the reviewers for their comments and suggestions. Below, we respond to each point raised. Following the helpful feedback from reviewers, we have addressed all comments, including (1) extensive analysis investigating and clarifying the selectivity metric, (2) analysis of train-test sequence similarity, preparing a new homology-filtered test set, (3) demonstration of CleaveNet's generalizability through *in silico* conditional design of selective substrates for MMP9, (4) benchmarking of the CleaveNet Predictor to the best-in-class substrate classification tool, and (5) additional rationale and clarification for the use of uncertainty in our substrate prioritization. We believe these revisions support the strength and robustness of CleaveNet and have improved our manuscript overall. Our response to each comment is given in purple. Direct quotes from our revision are given in *italics*. We thank the reviewers again for their constructive feedback.

Reviewer #1

This is an excellent, technically rigorous study that will be valuable to groups designing protease-responsive probes. The authors present CleaveNet, a two-part end-to-end AI pipeline tailored specifically for designing 10-mer peptide substrates targeting matrix metalloproteinases (MMPs). The CleaveNet Generator creates novel substrates, optionally conditioned on desired protease-cleavage profiles, while the CleaveNet Predictor evaluates cleavage efficiency and selectivity *in silico*. Both models were trained on extensive mRNA-display screening data from Kukreja et al. (2015), enabling careful uncertainty estimation via ensemble predictions. The comprehensive validation (including an extensive and informative 95-substrate *in vitro* screen focused on MMP-13) is particularly strong. Below, we highlight points that, if clarified or addressed through small additional analyses, could further bolster the manuscript's already substantial contributions.

We thank the reviewer for their careful assessment of our manuscript and for their excitement about the concept, broad utility, and novelty of our work.

Major comment #1:

Cross-protease selectivity distortion from Z-score normalization

The manuscript relies heavily on Z-scores to quantify cleavage efficiency, compute selectivity, and condition the generative model. However, because Z-scores were computed independently for each protease (i.e., normalized using each protease's own mean and standard deviation) differences in underlying variance (σ) across proteases may introduce artifacts when comparing Z-scores across MMPs. This is particularly relevant for CleaveNet's selectivity metric, which depends on differences in Z-scores across proteases.

To illustrate clearly, consider a simple numeric example involving two hypothetical proteases (A and D):

- Protease A cleaves substrates x, y, and z with raw cleavage values of 70, 13, and 17, respectively (mean = 33.3, $\sigma = 31.8$).
- Protease D cleaves the same substrates with raw cleavage values of 98, 1, and 1 (mean = 33.3, $\sigma = 56.0$).

Substrate x receives an identical Z-score (1.15) from both Protease A and Protease D.

Consequently, if selectivity scores are computed by subtracting the average Z-score of all other proteases (as defined in the manuscript), substrate x would yield very similar selectivity scores for both proteases, despite the stark differences in absolute cleavage levels and true biological selectivity. Protease D cleaves x far more selectively in absolute terms (raw = 98 vs. 1 and 1 for others), while A cleaves x more moderately and has a significant cleavage signal for the two other substrates (raw = 70 vs. 13 and 17 for others). Because σ is larger for D (56.0) than for A (31.8), this real difference in selectivity is flattened.

Conversely, the same mechanism could also inflate apparent selectivity if σ is small.

To be clear, this limitation affects cross-protease comparisons only. Within a single protease, Z-scores remain perfectly valid for ranking or classification. But for metrics like selectivity—which are computed from Z-score differences across proteases—variance heterogeneity may introduce distortions, potentially inherited by both CleaveNet’s conditioning and evaluation procedures. We recognize that the authors used Z-scores directly from Kukreja et al. (Chem. Biol. 2015), who applied a residual-based LOESS normalization to sequencing count data. While raw cleavage values were not made public, we believe it is reasonable to ask whether they can be requested from the original authors. We encourage the authors to:

- Reflect on how inter-protease differences in variance of raw cleavage data (σ) may bias selectivity scores.
- If feasible, attempt to obtain the raw data from Kukreja et al. to evaluate this directly. Alternatively, consider mitigation strategies such as estimating pseudo-raw cleavage magnitudes ($Z \times \sigma$) or adopting rank-based selectivity metrics that are more robust to scaling artifacts. At minimum, a short discussion of these limitations would help readers interpret CleaveNet’s selectivity outputs more cautiously and transparently.

We thank the reviewer for this important comment. Before delving into this discussion, we would like to clarify that “selectivity” in our manuscript refers to a given substrate, where high selectivity means that this substrate is preferentially cleaved by one MMP relative to all other MMPs. This is distinct from the “specificity” of an MMP, where high specificity indicates that an MMP cleaves a

very narrow set of substrates, compared to the full substrate set, which could potentially result in highly skewed cleavage distributions or high variance. We have now clarified this in the **Introduction** in lines 25-28 as follows:

*"Yet, designing substrates that are both **efficient—i.e., having high absolute cleavage—and selective—i.e., preferentially cleaved by a target protease over others**—remains a significant challenge rooted in the complex biochemistry of proteases (16, 17) and their substrates (18-20)."*

As the reviewer astutely pointed out, the selectivity metric defined by Kukreja et al., 2015 and utilized in our manuscript is imperfect and complicated by the fact that the data utilized to compute it is Z-scored. We reached out to Dr. Strongin to request the raw data from Kukreja et al. but unfortunately did not receive a reply. After Z-scoring, information on the absolute cleavage efficiencies that could be derived from raw sequencing reads from mRNA display is lost, making it challenging to compare across MMPs. Importantly, this limitation is independent of the specificity of a given protease across sequences in the library, which merely shifts the skewness and increases the σ of the cleavage distribution used to compute the Z-scores. More fundamentally, given the nature of mRNA display, true cleavage efficiencies and Z-score values may be calibrated very differently across MMPs. For example, a Z-score of 2 may correspond to 50 mRNA-display sequencing reads for MMP1 versus 1000 mRNA-display sequencing reads for MMP3, but these would be treated equivalently in our selectivity calculation. Furthermore, the unimodal cleavage Z-score distributions from Kukreja et al. do not clearly distinguish "cleaved" versus "non-cleaved" sequences.

Given these considerations and the unavailability of absolute raw read data, using the existing selectivity metric was the soundest choice for development and validation of our models. Despite this metric being suboptimal, this did not negatively affect the CleaveNet conditioning strategy for MMP13, relative to the unconditional generation and site-independent baselines (**Fig. 6**). Even if seed sequences used to generate the conditioning tag were derived from the top 50 most selective MMP13 substrates (with selectivity defined by this metric), CleaveNet-generated sequences achieved much higher *in vitro* selectivity than control groups and yielded new sequences that were uniquely cleaved by MMP13, attaining comparable *in vitro* selectivity to the mRNA display selective controls (**Fig. 7C**). We anticipate this being due to the small set of 50 seed sequences selected for conditioning. While this imperfect selectivity metric may fail to appropriately condition models when aiming for sequences with more average Z-scores, it is still likely to perform robustly when aiming for very high selectivity sequences for each protease, especially when conditioning on a small set ($n=50$) of sequences at the extreme of the cleavage distribution and thus de-risking our conditioning approach.

To the reviewer's point about cleavage magnitudes, we note that, relative to mRNA-display screens, *in vitro* fluorogenic screens enable more confident differentiation of cleaved versus non-cleaved substrates. In fluorogenic assays, cleaved substrates will exhibit a monotonic increase in signal over time and clearly separate from non-cleaved substrates. In contrast, true cleavage is harder to ascertain from sequencing data from mRNA-display screens, which are limited by being single timepoint, by sequencing errors, and by low-level contamination of non-cleaved barcodes. Since we detected cleavage events for all MMPs in the fluorogenic screen in our manuscript, we were able to infer approximate cleavage thresholds for each screened MMP (**Table S6, Fig. S23**). These thresholds reflect the Z-score values, predicted by the CleaveNet Predictor, under which the ROC-AUC for classifying cleaved versus non-cleaved was maximized given the true *in vitro* cleavage data. Critically, we note that computing these thresholds requires the ground truth raw *in vitro* cleavage data where it is possible to distinguish cleaved versus non-cleaved substrates, as in our *in vitro* screen.

Figure R1. mRNA Z-score distributions for MMPs in *in vitro* fluorogenic screen with inferred cleavage thresholds (red lines). Cleavage thresholds correspond to values for which ROC-AUC was maximized given true cleavage data collected *in vitro*.

Considering the reviewer's comment, we more closely inspected the cleavage scores and thresholds from our *in vitro* fluorogenic screen (**Figure R1**). Cleavage thresholds vary widely across MMPs (from 0.3 for MMP9 to 2.4 for MMP1), supporting the inferred mismatch between predicted Z-scores reported by Kukreja et al. and true cleavage efficiency between different MMPs (**Figure R1**). For instance, a Z-score of 2 would correspond to no cleavage by MMP1, but a top percentile cleavage for MMP12. Such differences could indeed confound the selectivity metric defined by Kukreja et al; again, we note that this validation was only enabled by our fluorogenic

screen data where we had a clear notion of cleaved versus non-cleaved. In our initial submission, we had proposed that users could take advantage of these cleavage thresholds to get more realistic metrics of cleavage efficiency across different proteases; however, we had not incorporated any methods supporting a correction to the selectivity metric.

To overcome this limitation and aid in data interpretation, in our revision we introduce new **Methods** that leverage cleavage thresholds to transform predicted Z-score values to "corrected efficiency" and "corrected selectivity" values, respectively. The new text on lines 872-898:

"Cleavage threshold selection. *Cleavage thresholds reflect the Z-score values, predicted by the CleaveNet Predictor, under which the ROC-AUC for classifying cleaved versus non-cleaved was maximized given the true in vitro cleavage data for the 95 substrates screened (Table S6). Lack of commercial options to purchase 6/18 MMPs, namely MMP11, MMP15, MMP16, MMP19, MMP24, MMP25, meant that the average cleavage threshold across the 12 MMPs screened (0.983) was imputed for these 6 MMPs."*

"Calculation of corrected efficiencies from predicted Z-score data. *Predicted Z-scores can be transformed to "corrected efficiency" values using the `corrected_efficiency` function, provided in the `Corrected efficiencies and specificities` notebook, that leverages cleavage thresholds inferred from the in vitro screen. To achieve this, MMP-specific cleavage thresholds are first subtracted from Z-scores, such that adjusted values that are positive correspond with true cleavage while negative values correspond to no cleavage. Next, adjusted values are transformed to corrected efficiency values according to the equation below. After said transformation, the sequence with the highest adjusted Z-score for a given MMP has a value of 1, a fractional efficiency between 0 and 1 is assigned to all other cleaved substrates, and a value of 0 is assigned to non-cleaved substrates. s , m , $Z_{s,m}$, and T_m correspond to a substrate, an MMP, a Z-score, and a cleavage threshold, respectively."*

$$\text{Corrected Efficiency } (\hat{E}_{s,m}) = \begin{cases} \frac{\hat{Z}_{s,m} - T_m}{(\hat{Z}_{s,m} - T_m)_{\max}}, & \hat{Z}_{s,m} > T_m \quad \text{(Cleaved)} \\ 0, & \text{otherwise (Non-cleaved)} \end{cases}$$

"Calculation of corrected selectivity scores. *Given that corrected efficiency scores bring all MMPs back to the same playing field with values between 0 and 1, relative cleavage differences between substrates can be more effectively compared. The `corrected_selectivity` function, provided*

in the *Corrected efficiencies and specificities notebook*, leverages corrected efficiency values to calculate corrected selectivity values according to the equation below. s , m , and E_{sm} correspond to a substrate, an MMP, and a corrected efficiency, and M is the total number of MMPs over which the selectivity score is being computed. "

$$\text{Corrected Selectivity } (CS_{S_m}) = \hat{E}_{s_m} - \frac{\sum_{i=0}^{i=M, i \neq m} \hat{E}_{s_i}}{M - 1}$$

We also include the following to the **Discussion** on lines 458-462:

"Our fluorogenic screen also enabled identification of cleavage thresholds that can be used to calibrate predicted Z-scores to true cleavage efficiencies for individual MMPs and to better compare relative cleavage across MMPs. Such thresholds, utilized by provided functions for correcting efficiency and selectivity scores, will greatly improve data interpretability, substrate selection, and model conditioning (Fig. S25)."

New **Figure S25** exemplifies differences between mRNA Z-scores and corrected efficiencies, highlighting improved data interpretation with the latter. Importantly, since correction is used for substrate selection or data interpretation, models do not need to be retrained.

sequence	MMP1	MMP2	MMP3	MMP7	MMP8	MMP9	MMP10	MMP11	MMP12	MMP13	MMP14	MMP15	MMP16	MMP17	MMP19	MMP20	MMP24	MMP25
LAAYRLEDTF	0.11	-0.52	0.18	1.28	0.07	0.66	0.91	-0.15	0.53	-0.05	-0.47	-0.11	-1.24	0.67	0.53	1.35	-0.38	0.55
QKARLIMQAI	0.2	0.36	1	1.34	0.13	1.59	0.27	0.17	0.54	-0.41	-0.38	-0.29	-0.44	0.85	1.66	0.51	-0.48	0.73
AARPGFGLSP	-0.85	-0.37	-0.73	-1.19	-1.26	-0.5	-1.32	-1.53	-1.92	-0.44	-0.68	-1.29	-0.94	-1.28	-0.65	-0.76	-0.99	-0.5
LKAYKSELEE	-0.57	-0.11	-0.12	0.6	-0.41	0.66	-0.9	-0.49	0.32	-0.38	-1.22	-0.93	-0.99	-0.44	-0.36	0.72	-0.86	-0.67
RTRDRLDEVK	0	-1.19	-0.46	-0.25	-0.05	-0.36	-0.18	-0.86	-0.32	-0.34	-0.28	-0.12	-0.71	0.04	0.83	0.02	-0.28	-0.08
RQWAGLVEKV	0.25	0.19	0.1	1.15	-0.23	0.49	-0.49	0.15	-0.03	-0.7	1.84	1.59	1.39	-0.23	0.63	0.86	1.12	-0.19
KGEYRTPED	-0.87	-1.19	-1.13	-0.82	-0.62	-0.98	-0.54	-0.49	0.23	-1.43	-0.38	-0.42	-0.72	-0.03	-0.7	0.27	-0.71	-0.48
STSGGYFYT	-1.25	-0.07	-0.32	-0.6	-0.37	-0.46	-0.82	-0.6	-0.94	-1.16	-0.22	-0.28	-0.28	-0.96	0	-1.31	-0.34	0.25
HIDDKAFENV	-0.47	-0.18	0.14	-0.06	-0.43	-0.71	-0.47	-0.87	-0.39	-0.61	-0.31	-1.03	-0.83	-0.57	0.07	0.1	-0.83	-0.51
TVNENLENY	-0.4	0.35	0.4	0	0.76	1.46	-0.36	0.3	1.72	-0.06	0.55	0.83	0.33	0.89	1.99	1.14	0.46	1.84
Threshold	2.4	1.0	1.0	0.9	1.3	0.3	0.8	1.0	0.4	1.0	1.0	1.0	1.0	1.0	1.0	0.7	1.0	1.0

sequence	MMP1	MMP2	MMP3	MMP7	MMP8	MMP9	MMP10	MMP11	MMP12	MMP13	MMP14	MMP15	MMP16	MMP17	MMP19	MMP20	MMP24	MMP25
LAAYRLEDTF	0.00	0.00	0.00	0.17	0.00	0.09	0.03	0.00	0.05	0.00	0.00	0.00	0.00	0.00	0.00	0.26	0.00	0.00
QKARLIMQAI	0.00	0.00	0.00	0.19	0.00	0.33	0.00	0.00	0.05	0.00	0.00	0.00	0.00	0.00	0.29	0.00	0.00	0.00
AARPGFGLSP	0.00	0.00	0.00	0.00	0.00	0.00	0.00	0.00	0.00	0.00	0.00	0.00	0.00	0.00	0.00	0.00	0.00	0.00
LKAYKSELEE	0.00	0.00	0.00	0.00	0.00	0.09	0.00	0.00	0.00	0.00	0.00	0.00	0.00	0.00	0.00	0.01	0.00	0.00
RTRDRLDEVK	0.00	0.00	0.00	0.00	0.00	0.00	0.00	0.00	0.00	0.00	0.00	0.00	0.00	0.00	0.00	0.00	0.00	0.00
RQWAGLVEKV	0.00	0.00	0.00	0.11	0.00	0.05	0.00	0.00	0.00	0.00	0.14	0.23	0.11	0.00	0.00	0.06	0.03	0.00
KGEYRTPED	0.00	0.00	0.00	0.00	0.00	0.00	0.00	0.00	0.00	0.00	0.00	0.00	0.00	0.00	0.00	0.00	0.00	0.00
STSGGYFYT	0.00	0.00	0.00	0.00	0.00	0.00	0.00	0.00	0.00	0.00	0.00	0.00	0.00	0.00	0.00	0.00	0.00	0.00
HIDDKAFENV	0.00	0.00	0.00	0.00	0.00	0.00	0.00	0.00	0.00	0.00	0.00	0.00	0.00	0.00	0.00	0.00	0.00	0.00
TVNENLENY	0.00	0.00	0.00	0.00	0.00	0.30	0.00	0.00	0.48	0.00	0.00	0.00	0.00	0.00	0.43	0.17	0.00	0.20

"Figure S25. Comparison between mRNA display Z-scores (top) and corrected efficiencies (bottom) for 10 sequences in the training set. Corrected efficiencies were calculated using the `corrected_efficiency` function that leverages cleavage thresholds inferred from the in vitro screening data (top table, last row italics). After transformation to corrected efficiencies, the sequence with the highest cleavage for a given MMP has a value of 1, a fractional efficiency between 0 and 1 is assigned to all other cleaved substrates, and a value of 0 is assigned to non-cleaved substrates. This improves data interpretability, substrate selection, and model conditioning. It also brings all values for all MMPs between 0 and 1, such that relative differences across substrates can be more

effectively compared within a given MMP. The `corrected_selectivity` function then streamlines selectivity analysis.”

Finally, we make available on the GitHub repository a Jupyter notebook titled “`Corrected efficiencies and specificities notebook.ipynb`” that contains two new functions, namely `corrected_efficiency` and `corrected_selectivity`, to streamline the use of these new metrics.

Major comment #2:

Generalisation beyond MMP-13 and Benchmarking: CleaveNet is framed as a family-wide design platform, yet all experimental validation targets MMP-13. Even a small in-silico case study on a biologically distinct enzyme (e.g. MMP-9 or MMP-14)—generating 10–20 candidate substrates, scoring them with the Predictor, and showing predicted selectivity—would make the generalisation claim more credible. Specifically, the authors could demonstrate that CleaveNet reliably shifts predicted substrate performance above random peptide baselines and confirm sequence novelty relative to training peptides.

We thank the reviewer for these suggestions on generalization. As the reviewer points out, our experimentally validated design task was focused on MMP13 targets, to maximize validation across substrates for both efficiency and selectivity, which was most feasible when focusing on a single protease target. We highlight that our work includes extensive *in silico* evaluations across other MMPs in the supplement (**Table S1, Figure S2-S12**) and furthermore correlated *in vitro* efficiency scores with CleaveNet predictions for the 95 tested substrates across 12 MMPs tested, including 11 MMPs beyond MMP13 (**Figure S23**).

However, we agree that the manuscript could certainly be strengthened by providing new evidence that the CleaveNet pipeline for selectivity prediction and selectivity-guided design generalizes to other MMPs, and that it can shift predicted substrate performance above baselines. To directly address the reviewer’s suggestion, we conducted new conditional generation and analysis for MMP9, analogous to the analysis of **Figure 6B** but for a biologically distinct enzyme. Importantly, MMP9 plays direct functional roles in multiple hallmarks of cancer, including angiogenesis, and has been a challenging target for the design of selective substrates, due to the high overlap of substrate recognition with MMP2, both of which are closely related gelatinases (Ratnikov et al. 2014).

In this new evaluation, we designed 20,000 candidate MMP9-selective substrates using CleaveNet’s pipeline for conditional generation with a conditioning tag specifying high MMP9-selectivity. We also sampled 20,000 sequences from the distribution of the top 50 MMP9-selective

sequences for the conditional site-independent baseline. We then predicted MMP9-selectivity scores and benchmarked against the mRNA-display dataset sequences, unconditional CleaveNet generations, and the site-independent baselines as was done in **Figure 6B**. These new results are included as the new **Figure S17**, reproduced below, and demonstrate that CleaveNet's conditionally-generated sequences achieve significantly greater selectivity scores than both unconditional generations and samples from the site-independent baselines, indicating that CleaveNet can reliably shift predicted substrate performance (**Figure S17**). We believe this *in-silico* evaluation of design for MMP9 selectivity supports the generalizability of CleaveNet to other MMPs.

Figure S17. Evaluation of substrates conditionally generated for high MMP9 selectivity. Predicted MMP9-selectivity scores for sequences conditionally generated by CleaveNet for high MMP9 selectivity ($n=20,000$, dark green), relative to sequences from the mRNA-display dataset ($n=18,583$, blue), unconditional generations ($n=19,905$, light green), and unconditional and conditional site-independent baselines ($n=20,000$ each, light and dark yellow, respectively). All pairwise comparisons are significant ($p < 0.0001$) via Kruskal-Wallis test.

To assess sequence novelty compared to the training data, we calculated the Levenshtein distance between all training sequences and the 20,000 CleaveNet-generated substrates conditionally designed for MMP9 selectivity. No exact matches were identified between the generated sequences and those in the training set. Additionally, for the top 10 MMP9 selective sequences generated by CleaveNet, we identified their closest matches in the training data, identified by the sequence with the smallest computed Levenshtein distance (**Table S5**). The top 10 sequences all differed between 4 to 6 residues from their nearest matches in the training set, indicating sufficient sequence novelty.

Table S5. Summary of sequence novelty for substrates generated in silico for MMP9 selectivity. Closest sequence match in the training set, determined by Levenshtein distance, to the top MMP9 selective substrates, conditionally designed by CleaveNet.

Conditionally generated sequence	Closest training match	Levenshtein distance
VVVI AVLQIM	VTVIALLRGQ	5
VQFVAVMSTV	VFVRALISTG	5
PLPFAGVAFM	PFWFALVAKG	5
FLPLGVGVAL	FYPRNIGVAL	4
QGPVVVGLA	SGPQAVVKTA	5
VMVRVTMVMV	VMVRFLMNQQ	5
MFVMGVFATV	PRVMLLFATG	5
LVVPAVVVMI	LVVAALVNVI	4
AGVVMLVGIM	SGPVMLRGTA	5
FPKMGVMGLV	PKMSHLRGGV	6

We have added the new **Figure S17 and Table S5** to the manuscript and describe these new results in the manuscript text as follows on lines 362-369:

“To demonstrate the generalizability of our approach, we used CleaveNet to design substrates selective for MMP9, a biologically-distinct enzyme that plays direct functional roles in multiple hallmarks of cancer and has been a challenging target for the design of selective substrates, due to high overlap of substrate recognition with MMP2 (Ratnikov et al. 2014). CleaveNet’s conditionally-generated sequences achieved significantly greater in silico MMP9-selectivity scores than both unconditional generations and samples from the site-independent baselines (Figure S17) and demonstrated substantial sequence novelty from the training set (Table S5), supporting the generalizability of the CleaveNet pipeline to multiple MMPs.”

Additionally, a brief benchmarking comparison (if feasible) against established protease predictors (e.g., DeepCleave, PROSPERous, other) would help contextualize CleaveNet’s predictive accuracy within the current state-of-the-art. If direct benchmarking is not feasible, clarifying why this is the case would strengthen the manuscript by clearly delineating CleaveNet’s scope and comparative advantages.

We thank the reviewer for this comment and agree that including an external benchmark would strengthen and delineate CleaveNet’s advantages. To our knowledge, CleaveNet is the only available tool that is capable of returning continuous cleavage efficiencies for protease substrates.

However, other binary cleavage prediction tools exist. Currently, ProsperousPlus (Li et al., 2023) appears to be the most comprehensive tool available for predicting cleavage events for MMPs. While it is applicable to a wider range of protease families, its predictive capabilities are currently limited to only seven human metalloproteases: MMP1, MMP2, MMP3, MMP7, MMP8, MMP9, and MMP14.

To benchmark ProsperousPlus against CleaveNet, we did the following, also added as a new **Method** on lines 900-909:

“Comparing the cleavage classification performance between CleaveNet and state-of-the-art ProsperousPlus. Classification performance (identification of cleaved versus non-cleaved) was compared for the 95 substrates we tested in vitro across the 7 MMPs available on ProsperousPlus (MMP1, MMP2, MMP3, MMP8, MMP9, MMP7, and MMP14). To classify substrates using CleaveNet, substrates were assigned a cleaved label if their predicted Z-score from the CleaveNet Predictor (transformer) was greater than its corresponding cleavage threshold in Table S6. ProsperousPlus was also used to assign a cleavage probability. Since ProsperousPlus does not specify cleavage cutoffs for individual MMPs, the default cleavage cutoff of 0.5 was used for this analysis. Sensitivity, specificity, and accuracy for individual MMPs were computed.”

Table R1 summarizes raw true positive, true negative, false positive, and false negative counts and **Table S9** (also included in the **Supplementary Information**) summarizes classification metrics for each model.

While ProsperousPlus achieved comparable sensitivity for cleavage to CleaveNet (average sensitivity of 91.3+/-15.5% vs. 90.7+/-4.49% for ProsperousPlus and CleaveNet, respectively), it demonstrated substantially lower specificity values (22.2+/-13.2% vs. 88.2+/-14.6% for ProsperousPlus and CleaveNet, respectively) and thus resulted in lower accuracy than CleaveNet (51.9+/-9.1% vs. 90.7+/-5.1% for ProsperousPlus and CleaveNet, respectively). The numerous false positives from ProsperousPlus matched or outnumbered the number of true positives, making the tool worse than the flip of a coin at identifying cleaved substrates. These results support the superiority of CleaveNet over the current most comprehensive tool, ProsperousPlus, for substrate cleavage classification. The following text was added to the **Discussion** on lines 474-476:

“Additionally, when used in isolation, the CleaveNet Predictor outputs continuous cleavage scores, as opposed to binary cleavage labels, **and even outperforms state-of-the-art tools for binary substrate classification (Table S9).**”

Table R1. Number of true positives (TP), true negatives (TN), false positives (FP) and false negatives (FN) for 95 substrates from *in vitro* screen using ProsperousPlus vs. CleaveNet.

Protease	ProsperousPlus		CleaveNet		ProsperousPlus		CleaveNet	
	TP	TN	TP	TN	FP	FN	FP	FN
MMP1	26	14	23	68	55	0	1	3
MMP8	24	27	36	54	27	17	0	5
MMP9	62	4	59	20	29	0	13	3
MMP7	27	15	31	60	48	5	2	2
MMP2	44	8	42	39	43	0	12	2
MMP3	43	6	36	50	46	0	2	7
MMP14	31	14	29	56	49	1	7	3

“Table S9. Classification performance comparison between ProsperousPlus and CleaveNet. Both tools were utilized to classify (cleaved vs. non-cleaved) the 95 substrates used in the in vitro fluorogenic screen against the 8 MMPs available in ProsperousPlus. While ProsperousPlus achieved comparable sensitivity for cleavage to CleaveNet (average sensitivity of 91.3+/-15.5% vs. 90.7+/-4.49% for ProsperousPlus and CleaveNet, respectively), it demonstrated substantially lower specificity values (22.2+/-13.2% vs. 88.2+/-14.6% for ProsperousPlus and CleaveNet, respectively) and thus resulted in lower accuracy than CleaveNet (51.9+/-9.1% vs. 90.7+/-5.1% for ProsperousPlus and CleaveNet, respectively).”

Protease	ProsperousPlus	CleaveNet	ProsperousPlus	CleaveNet	ProsperousPlus	CleaveNet
	Sensitivity		Specificity		Accuracy	
MMP1	100	88.5	20.3	98.6	42.1	95.8
MMP8	58.5	87.8	50	100	53.7	94.7
MMP9	100	95.2	12.1	60.6	69.5	83.2
MMP7	84.4	93.9	23.8	96.8	44.2	95.8
MMP2	100	95.5	15.7	76.5	54.7	85.3
MMP3	100	83.7	11.5	96.2	51.6	90.5
MMP14	96.9	90.6	22.2	88.9	47.4	89.5
Average	91.3	90.7	22.2	88.2	51.9	90.7

Major comment #3:

Train–test sequence similarity.

The paper says the 18 583-peptide dataset was split 80/20 at random (Methods, p 40), so peptides differing by only a few residues may appear in both splits. Because short 10-mers can share identical stretches, this can over-estimate predictive performance. We recommend re-running the

split with a simple clustering filter (e.g., CD-HIT at 70 % identity or removing any train–test pair with Levenshtein distance < 3) and reporting the resulting MAE/ROC. Even if the overall ranking is unchanged, showing robustness to a similarity-controlled split would strengthen the generalisation claim. (The external 71-peptide fluorescence set is a helpful check, but its limited size and coverage make the within-dataset control important as well.)

This is an excellent suggestion by the reviewer. We quantified possible overlap between the mRNA display train and test set sequences by computing the Levenshtein distance between each test set sequence and every train sequence. We determined the closest training match to a test sequence by taking the minimum of these distances. From this, we determined that 816 of the 3,717 sequences (from the original mRNA-display test set) had a distance less than 3 to at least 1 sequence in the training set, indicating possible homologous overlap.

We prepared a new test set by removing these 816 sequences from the original test set, resulting in a new homology-filtered mRNA-display test set comprised of 2,901 sequences, that do not overlap with the training data. We redid all the relevant analysis for the mRNA display test set, using this new homology-filtered mRNA test set, and overall, we observed nearly negligible changes in model performance, confirming the models' generalizability.

First, we compared the MAE per-MMP for both the old random test set and new homology-filtered test set, showing minimal performance change for both the LSTM and Transformer models. The Transformer's error remains within the ensemble standard deviation (**Table R2**), while the LSTM displays slightly more variance but similar MAE values across sets (**Table R3**).

Table R2. Comparison of ensemble-averaged MAE scores from the CleaveNet Transformer Predictor on the random mRNA-display test set versus the homology-filtered mRNA-display test set. For each MMP the mean average error (MAE) and the standard deviation over an ensemble of n=5 trials is reported. The absolute difference between the average values in both columns is also reported.

	Random mRNA-display test	Homology-filtered mRNA-display test	Absolute difference in MAE
MMP1	0.427 ± 0.004	0.421 ± 0.006	0.006
MMP2	0.479 ± 0.007	0.478 ± 0.007	0.001
MMP3	0.548 ± 0.007	0.546 ± 0.007	0.002
MMP7	0.529 ± 0.009	0.534 ± 0.008	0.005
MMP8	0.489 ± 0.006	0.491 ± 0.006	0.002
MMP9	0.580 ± 0.008	0.583 ± 0.008	0.003
MMP10	0.512 ± 0.004	0.512 ± 0.003	0.000

MMP11	0.466 ± 0.003	0.462 ± 0.005	0.004
MMP12	0.543 ± 0.007	0.554 ± 0.007	0.001
MMP13	0.459 ± 0.006	0.468 ± 0.004	0.009
MMP14	0.498 ± 0.005	0.498 ± 0.007	0.000
MMP15	0.485 ± 0.005	0.488 ± 0.007	0.003
MMP16	0.479 ± 0.005	0.478 ± 0.006	0.001
MMP17	0.514 ± 0.004	0.513 ± 0.006	0.001
MMP19	0.625 ± 0.004	0.620 ± 0.004	0.005
MMP20	0.535 ± 0.004	0.538 ± 0.003	0.003
MMP24	0.505 ± 0.004	0.504 ± 0.005	0.001
MMP25	0.558 ± 0.006	0.560 ± 0.008	0.002

Table R3. Comparison of ensemble-averaged MAE scores from the CleaveNet LSTM predictor on the random mRNA-display test set versus the homology-filtered mRNA-display test set. For each MMP the mean average error (MAE) and the standard deviation over an ensemble of n=5 trials is reported. The absolute difference between the average values in both columns is also reported.

	Random mRNA-display test	Homology-filtered mRNA-display test	Absolute difference in MAE
MMP1	0.454 ± 0.007	0.452 ± 0.012	0.002
MMP2	0.512 ± 0.004	0.518 ± 0.002	0.006
MMP3	0.555 ± 0.007	0.561 ± 0.002	0.006
MMP7	0.533 ± 0.004	0.545 ± 0.002	0.012
MMP8	0.516 ± 0.005	0.519 ± 0.009	0.003
MMP9	0.587 ± 0.005	0.599 ± 0.003	0.012
MMP10	0.521 ± 0.003	0.529 ± 0.003	0.008
MMP11	0.469 ± 0.005	0.448 ± 0.010	0.021
MMP12	0.548 ± 0.006	0.562 ± 0.009	0.014
MMP13	0.494 ± 0.004	0.514 ± 0.005	0.020
MMP14	0.526 ± 0.006	0.522 ± 0.005	0.004
MMP15	0.509 ± 0.006	0.515 ± 0.002	0.006
MMP16	0.500 ± 0.003	0.502 ± 0.003	0.002
MMP17	0.541 ± 0.003	0.547 ± 0.006	0.006
MMP19	0.624 ± 0.003	0.629 ± 0.002	0.005
MMP20	0.557 ± 0.008	0.567 ± 0.008	0.010
MMP24	0.525 ± 0.005	0.526 ± 0.003	0.001
MMP25	0.573 ± 0.001	0.579 ± 0.003	0.006

Further, when averaging MAE across MMPs, to understand the impact of the test set in the MAE analysis presented in **Figure 2B**, we again observe a minimal change to these scores (**Table R4**). **Table R4. Comparing MMP-averaged MAE for the random vs. homology-filtered test set.**

Model architecture	Random mRNA Test	Homology-filtered mRNA Test
Transformer	0.513 ± 0.048	0.514 ± 0.048
LSTM	0.530 ± 0.041	0.536 ± 0.043

Lastly, we observe a similar negligible difference in AUC performance. Below, we demonstrate this in a side-by-side comparison of the ROC-AUC plot for MMP13 on the old random test set alongside the new results for the homology-filtered test set (**Figure R2**), noting negligible differences in the computed AUC with the new filtered test set. We recomputed these on the homology-filtered test for the Transformer (**Figure S4**) and LSTM (**Figure S5**) model and find that these results are consistent across all MMPs.

Figure R2. ROC-AUC plot evaluating the Transformer predictor. Measured over the old random mRNA display test set (left) and the new homology-filtered mRNA display test set (right).

For completeness, and to emphasize model robustness, in the manuscript we replace all analyses computed using the random test split with the new homology-filtered mRNA-display test set. Namely, we updated the following figures and tables in the manuscript: **Figure 2B, C, and D**; and results for the mRNA-display test set in **Table S1; Figure S2, S3, S4, and S5**. We reproduce these figures below and describe these changes for reference. Lastly, we also upload the new filtered test splits in the github repository under `splits/kukreja` as `x_test_filtered.csv` and `y_test_filtered.csv`.

Figure 2 now includes updates to panels B, C, and D using the new homology-filtered test set, denoted here still as the mRNA-display test:

Figure 2. CleaveNet accurately predicts cleavage efficiencies of synthetic peptides against MMPs. Multi-output cleavage score regression and uncertainty estimation by the CleaveNet Predictor. **(A)** Two biochemically-distinct sets of synthetic peptides, from mRNA display ($n=2,901$ peptides) and fluorometric ($n=71$ peptides) activity assays, are used as test sets for evaluation across different MMPs. IceLogos are normalized by natural amino acid frequencies (see Methods). Individual amino acids are colored by the chemical properties of their side chain: hydrophobic aromatic (lavender), hydrophobic (blue), hydrophilic (yellow), acidic (orange), and basic (red).

In **Table S1** we replace the values in columns "Transformer mRNA-display Test" and "LSTM mRNA-display Test" with the MAE calculated using the new homology-filtered test. This is still denoted as mRNA-display Test:

Table S1. Performance of the CleaveNet Predictor. For each MMP and test set, the average mean absolute error (MAE) and standard deviation over an ensemble of $n=5$ is provided for each model class.

Protease	Transformer	LSTM	Transformer	LSTM
	mRNA-display Test	mRNA-display Test	Fluorescence Test	Fluorescence Test
MMP1	0.421 ± 0.006	0.452 ± 0.012	0.636 ± 0.058	0.513 ± 0.045
MMP2	0.478 ± 0.007	0.518 ± 0.002	-	-
MMP3	0.546 ± 0.007	0.561 ± 0.002	0.593 ± 0.034	0.601 ± 0.041
MMP7	0.534 ± 0.008	0.545 ± 0.002	0.764 ± 0.093	0.808 ± 0.089
MMP8	0.491 ± 0.006	0.519 ± 0.009	-	-
MMP9	0.583 ± 0.008	0.599 ± 0.003	-	-
MMP10	0.512 ± 0.003	0.529 ± 0.003	0.713 ± 0.050	0.652 ± 0.035
MMP11	0.462 ± 0.005	0.448 ± 0.010	-	-
MMP12	0.554 ± 0.007	0.562 ± 0.009	0.793 ± 0.058	0.801 ± 0.041
MMP13	0.468 ± 0.004	0.514 ± 0.005	0.610 ± 0.055	0.590 ± 0.045
MMP14	0.498 ± 0.007	0.522 ± 0.005	-	-
MMP15	0.488 ± 0.007	0.515 ± 0.002	-	-
MMP16	0.478 ± 0.006	0.502 ± 0.003	-	-
MMP17	0.513 ± 0.006	0.547 ± 0.006	0.595 ± 0.040	0.534 ± 0.046
MMP19	0.620 ± 0.004	0.629 ± 0.002	-	-
MMP20	0.538 ± 0.003	0.567 ± 0.008	-	-
MMP24	0.504 ± 0.005	0.526 ± 0.003	-	-
MMP25	0.560 ± 0.008	0.579 ± 0.003	-	-

In **Figure S2** and **Figure S3** we update plots to include only data for the homology-filtered test set, and recompute the correlation coefficient depicted in each plot:

Figure S2. Performance of the LSTM model on Z-score prediction over the mRNA-display test set for each MMP. The correlation coefficient (Pearson's r) is denoted for each plot.

Figure S3. Performance of the Transformer model on Z-score prediction over the mRNA-display test set for each MMP. The correlation coefficient (Pearson's r) is denoted for each plot.

In **Figure S4** and **Figure S5** we update plots to include only data for the homology-filtered test set, and recompute the AUC values depicted for each plot:

Figure S4. ROC-AUC plot evaluating the transformer predictor over the mRNA-display test set for each MMP. Individual lines represent performance for different Z-score thresholds, with AUCs provided.

Figure S5. ROC-AUC plot evaluating the LSTM predictor over the mRNA-display test set for each MMP. Individual lines represent performance for different Z-score thresholds, with AUCs provided.

We describe these new results in the manuscript text in lines 107-113 as follows:

*“To assess model performance and generalizability, we evaluated prediction performance across two datasets: sequences obtained from a 20% random split of the training dataset **that were further filtered for homology against the training dataset (see Methods) and** never seen during training (mRNA-display test) and sequences obtained from an independent set of 71 FRET-paired sequences that were previously screened against 7 recombinant MMPs in vitro (fluorescence test) (52).”*

We also expand the specifics on the mRNA-display test split and homology-based filtering in the **Methods** section under **Datasets**, with the new text below on lines 699-705:

“An 80/20 split was performed on the data to create training and test sets, resulting in 3,717 sequences in the test set. To assess overlap between the train and test datasets, the minimum Levenshtein distance between each test sequence and the training set was calculated. It was found that 816 test sequences closely matched the training set (distance <3). These sequences were removed, leaving 2,901 non-overlapping sequences as the homology-filtered test set. This set was held out from all models during training and is referred to as the mRNA-display test set.”

Major comment #4:

Uncertainty estimation and ensemble details.

The manuscript mentions that the CleaveNet Predictor is an ensemble of five models and that substrate selection is guided by an "uncertainty-aware cleavage score" ($\hat{Z}_{sm} - \sigma_{sm}$). However, it remains unclear how the uncertainty score σ_{sm} is computed (e.g., whether it is the standard deviation of the ensemble predictions), and whether it correlates meaningfully with true prediction error. We suggest clarifying the method used to derive σ_{sm} , briefly describing the differences among the five models (e.g., random seeds vs. data splits), and if possible adding a calibration figure or summary statistic showing how σ_{sm} relates to observed error on the test set. This would strengthen the rationale for using uncertainty in substrate triage and enhance reproducibility.

We thank the reviewer for the suggestion. The uncertainty score σ_{sm} is computed as the standard deviation of the predicted Z-scores from an ensemble of predictor models (n=5). We have clarified this with the following edits.

The method for computing the uncertainty score is now expanded on lines 99-102 where it is first introduced. The sentence now reads:

“Given an input amino acid sequence, s , the model predicts continuous cleavage values \hat{Z}_{sm} for 18 MMPs. Their associated uncertainty scores are quantified as the standard deviation of the predicted Z-scores measured by training an ensemble of 5 predictor models over the mRNA-display dataset.”

We have expanded the specifics on the uncertainty score in the **Methods** section **Training the CleaveNet predictors** on lines 732-735 with the new text below:

*“To quantify model uncertainty and performance, an ensemble (Lakshminarayanan 2017) of 5 predictor models was trained over 5 independent 80/20 train/validation splits of the original training dataset and evaluated using the independent mRNA display test set. **The uncertainty score is quantified as the standard deviation of the predicted Z-scores from each of these 5 models.**”*

Additionally, we have added a new figure depicting the relationship between the uncertainty metric and the observed absolute error for the CleaveNet Predictor across all MMPs, **Figure S10**. Here, observed error is computed as the absolute difference between predicted and true cleavage scores. Using a linear fit to understand the relationship between these two variables, we observe a positive slope for all MMPs, however with weak correlations given by low R^2 . In the edge cases, the model is sometimes overconfident with large errors. The uncertainty is thus only a coarse metric, representing the variance in model predictions, and not perfectly calibrated to the observed error. We expand upon these caveats in the manuscript and encourage the users to use this metric in combination with other filtering criteria. **Figure S10** is reproduced below and discussed in the manuscript as follows:

Figure S10. Uncertainty versus observed absolute error for CleaveNet Predictor on the mRNA-display test set sequences and across MMPs.

We describe the relationship between observed error and uncertainty in lines 143-149:

*“Independent of their exact value, we expect low Z-scores to be non-cleaved and higher scores to be cleaved. The CleaveNet Predictor consistently predicted lower predicted Z_{sm} for sequences that were not cleaved (blue) relative to substrates that were cleaved (red) (Fig. 2). **The uncertainty of the model coarsely correlated with the absolute error in predicted Z_{sm} (Fig. S10), and reflected the confidence of individual predictions, with relatively low uncertainty for most non-cleaved substrates or top cleaved substrates (Fig. 2G, Fig. S11, S12).***

Minor comment #1:

The text briefly notes that MMP-13 is implicated in cancer, wound healing and osteoarthritis, but doesn't say why its selectivity is unusually hard to achieve or therapeutically valuable. A short expansion in the Introduction or Results, e.g., noting its broad substrate overlap with other collagenases and current interest in disease-specific diagnostics, would ground the choice of showcase target.

We thank the reviewer for this suggestion, which we have incorporated into the **Discussion** on lines 446-450:

*"We validated CleaveNet by experimentally testing substrates designed for high MMP13 efficiency and selectivity in vitro against 12 recombinant MMPs. **This task, once exceedingly difficult due to the extensive substrate overlap of MMP13 with collagenases and gelatinases (Eckhard et al., 2016, Kukreja et al., 2015), remains essential for developing effective MMP13-specific profiling tools and MMP13-activated diagnostics and therapeutics.**"*

Minor comment #2:

Readers first encounter the 10-mer length midway through the Introduction. Consider moving that statement up (e.g., into the Abstract or Figure 1 caption) so it's clear from the outset that CleaveNet currently designs fixed-length 10-residue peptides, even though 8-mer substrates are common in the protease literature.

We thank the reviewer for raising this point, as we realize that the specifics around generation length could be clarified. The comment that CleaveNet designs fixed length 10-mer peptides is in fact not correct. The CleaveNet Generator model is autoregressive, so while the model is trained on 10-mers, a fixed sequence length is not explicitly enforced during training or during inference (i.e., generation). The model designs new substrates by predicting the next token, given the previous tokens, until it reaches a [STOP] token. This means that generated sequences can be shorter than, equal to, or longer than 10 residues. For the purposes of evaluation and controlled comparison, we utilized CleaveNet-generated sequences that were 10 residues in length, similar to the mRNA display substrates utilized as controls. We have added the following new text to the "Unconditional generations" section of the manuscript Methods lines 779-783:

"New tokens were predicted given the previous tokens until a [STOP] token was reached. Because of this, generated sequences could be shorter than, equal to, or longer than 10 residues in length. Generated sequences shorter or longer than 10 residues in length were filtered out for comparison purposes; this filtered out 87 sequences."

This is additionally described in the Results lines 159-162:

*"To generate sequences relevant to MMP activity, we trained an autoregressive transformer model, **which can generate variable-length sequences**, on the training split of the mRNA-display dataset (see Methods for details).*

Finally, we note that the mentions of 10-mer in the Introduction are solely for demonstration, i.e., *"for a 10-amino acid peptide, approximately 20^{10} (c.a. 10^{13}) sequences are possible"*; there is no mention that CleaveNet designs or predictions are restricted to 10 residues.

While we validate only on 10-mer sequences in the context of the CleaveNet Generator, we note that the CleaveNet Predictor is validated *in silico* on sequences as short as 7 residues in length and as long as 14 residues in length using the fluorescence test set. Guidance on using the CleaveNet Predictor to obtain predictions for shorter sequences, by implementing PAD tokens on both ends, is described in the GitHub repository. The length differences in the mRNA-display and fluorescence test sets, in the context of the CleaveNet Predictor, are described in the manuscript Results on lines 113-114:

"Substrates across the two test sets differed in length (10-mers vs. 7- to 14-mers for mRNA-display vs. fluorescence, respectively)..."

In vitro validation of shorter or longer sequences in future work would inform the quality of these generations. In sum, the CleaveNet Predictor and CleaveNet Generator can score and design peptides of other lengths, though designed peptides that are not 10-mers have yet to be validated *in vitro*.

State explicitly whether k-mer counts ignore position. If positional context is not tracked, add a one-line note so readers don't assume site-specific motifs are being compared.

We agree that the details surrounding k-mer counts were incomplete. The k-mer counts are in fact position independent, and we have updated the text accordingly. When the k-mer analysis is introduced, we now state the following on line 181:

*"To further evaluate the plausibility of generated sequences, we inspected the cumulative density function (CDF) of unique **position-independent** k-mers in each set of generated or mRNA-display sequences (Fig. 3F)."*

Minor comment #3:

The main text discusses DL42 as a top hit, whereas Table S4 lists DL52. Please identify which ID is correct and adjust the other location (or note that both are distinct examples).

We appreciate the reviewer making this observation and apologize for the confusion. The correct ID is DL52, as described in **Table S4**. We have updated the main text on line 315:

*"In contrast, DL73 (the top MMP13 substrate) and **DL52** shared at most a single 4-mer with a small subset of training sequences exhibiting variable Z-scores."*

Minor comment #4:

Table S1 and ROC curves cover all 18 MMPs, but only the LSTM has per-MMP scatter plots (Fig S2). Providing the same scatter panels for the transformer, or merging the two into a single comparative figure, would give readers symmetrical insight.

We thank the reviewer for this comment and agree that including the per-MMP scatter plots for the transformer predictor would give the readers additional insights. As such, we have included an additional new supplementary figure, **Figure S3**, that shows the per-MMP scatter plots for the transformer CleaveNet Predictor.

Figure S3: Performance of the Transformer model on Z-score prediction over the mRNA-display test set for each MMP. The correlation coefficient (Pearson's r) is denoted for each plot.

We have updated the manuscript text to reference and discuss these results on lines 116-119 and on lines 127-128:

“Both the LSTM and transformer models displayed similar predictive performance on the test sets, as supported by their comparable mean absolute errors (MAE) between the true and predicted cleavage scores (Fig. 2B, Table S1) and by the strong correspondence between predicted Z_{sm} and true Z_{sm} for both models on the mRNA-display test set (Fig. S2, S3).”

“We observed strong correspondence between predicted Z_{sm} and true Z_{sm} for MMP13 (Pearson's $r=0.80$; Fig. 2C) and for other MMPs in the mRNA-display test set (Fig. S3).”

Minor comment #5:

Align Figure 7A caption with what is shown. The caption references yellow and burgundy baseline groups that are visible only in Fig S15. Either embed Fig S15 into the main text or trim the colour references in Figure 7A's caption so it accurately reflects the standalone figure.

We thank the reviewer for pointing this out. We believe the colors presented in the caption of 7A are important for interpreting the plots in the remaining panels B, C, D, and E. As such, we included a reference to **Figure S15** (now **Figure S18**) to improve the clarity in the figure caption. The updated figure caption is reproduced below:

*"(A) Schematic overview of nomination strategies for substrates included in the in vitro screen (n=95 substrates total). (i) Substrates were selected from unconditional generations for efficient cleavage by MMP13 (top) or conditionally designed to be cleaved selectively by MMP13 (bottom). In addition to CleaveNet-generated substrates (green, n=24 per group), appropriate baselines consisting of site-independent baseline alone (yellow, n=8 per group) and site-independent + CleaveNet Predictor (burgundy, n=8 per group) were added; **a schematic of these baselines is depicted in Fig. S18**. Controls from the mRNA-display training set corresponding to substrates that were efficiently, selectively, or not cleaved by MMP13 (light blue, dark blue, and grey respectively, n=5 per group) were also included."*

Minor comment #6:

Substantiate "diverse" claim for screened peptides. A simple pair-wise similarity heat-map (e.g., Levenshtein distance or clustering dendrogram) for the 95 tested substrates would back up the statement that the experimental set is sequence-diverse.

We agree that this analysis would further support our statement about sequence diversity. In response, in a new analysis, we created a pairwise similarity map (of inverse Levenshtein distances) for all 95 substrates, where a value of 10 indicates identical sequences and 0 indicates no similarity, and included this as the new **Figure S19**. The mean similarity across the 95 tested substrates is 2.1, showing most sequences differ significantly; the maximum similarity observed between two sequences is 8. In an additional new analysis in **Figure S20**, we visualized the distribution of similarity scores and observed that most sequence pairs exhibit low similarity, with only 1.23% of pairs showing moderate similarity (>5) and just 0.07% showing high similarity (>7).

Figure S19. Pairwise similarity map of the 95 substrates tested in vitro. Heatmap showing pairwise sequence similarities calculated using Levenshtein distance on a 0-10 scale. Similarity is measured as Levenshtein distance subtracted from the sequence length. A value of 10 represents identical sequences and a value of 0 represents no shared residues between each pair. Each axis plots the 95 substrate sequences, colored by similarity scores.

Figure S20. Distribution of pairwise sequence similarities of the 95 substrates tested in vitro. Histogram showing the frequency distribution of similarity scores for all unique pairwise comparisons among the 95 substrates tested in vitro. The red dashed line indicates the mean similarity (2.11).

We have also updated the manuscript text to discuss these results on lines 378-380:

“The 95-substrate panel was highly diverse in sequence space, as evidenced by a mean pairwise sequence similarity of 2.1 across the 95 tested substrates (Fig. S19, S20).”

Minor comment #7:

Fully label the heat-map in Figure 7B (or add an SI version). Only 24 of 95 substrates are currently annotated, which makes it hard to trace individual examples (DL41, DL48, etc.). A fully labelled version—either in the main figure or in Supplementary Information—would greatly improve interpretability.

We agree with the reviewer that a fully annotated heatmap would improve interpretability. Given space restrictions in **Figure 7B**, we have added a new supplementary figure, **Figure S24**, of an enlarged and completely annotated heatmap of *in vitro* cleavage efficiencies for all substrate-protease pairs in **Fig. 7A**. This new figure is referenced in the manuscript on line 392:

“Visualizing the cleavage efficiencies of all protease-substrate pairs demonstrated that substrates designed for MMP13 efficiency were highly cleaved by MMP13 but also fairly promiscuously cleaved by other proteases, while substrates designed for MMP13 selectivity were more selectively cleaved by MMP13, consistent with design expectations (Fig. 7B, Fig. S24).”

Figure S24. Heatmap showing in vitro cleavage efficiencies for all substrate-protease pairs. Substrates were ordered by their expected MMP13 cleavage profile: efficient (top), selective (middle), uncleaved (bottom). In addition to ClaveNet-generated substrates (green, n=24 per group), appropriate baselines consisting of site-independent baseline alone (yellow, n=8 per group) and site-independent + ClaveNet Predictor (burgundy, n=8 per group) were added. Controls from the mRNA-display training set corresponding to substrates that were efficiently, selectively, or not cleaved by MMP13 (light blue, dark blue, and grey respectively, n=5 per group) were also included.

Comment on code:

very solid repository. The repository already meets reproducibility guidelines and should be a usable resource for other groups. small suggestion (sorry if it's already there and we missed it): Provide a small CSV of the 95 in-vitro peptides + raw cleavage values so users can reproduce Figure 7 directly.

We thank the reviewer for their very positive comments on the CleaveNet repository. We agree that including the raw data to reproduce **Figure 7** would be valuable. As suggested, we have included an excel sheet of the 95 *in vitro* peptides and their raw cleavage fold changes from time 0 and efficiencies in the Github data folder. The efficiency values are the values plotted in **Figure 7**; however, we include the raw cleavage values as well per the reviewer's request.

In vitro cleavage efficiencies and raw cleavage values have been uploaded to the GitHub repository: `data/invitro_efficiencies_7B.xlsx` and `data/Raw_in_vitro_cleavage_fold_changes.xlsx`, respectively.

Reviewer #2

Martin-Alonso et al. in publication „Deep learning guided design of protease substrates“ submitted to Nature Communications present an interesting *in silico* AI based approach to analyzing and predicting protease substrate specificity, focusing on modeling substrate–enzyme interactions. The methodology applied and the attempt to formulate a general predictive model are commendable, particularly in light of the growing importance of bioinformatics and AI in molecular biology.

However, the central research hypothesis—that substrate specificity can be reliably predicted using the proposed model—has not been convincingly validated. A critical limitation of the study is the exclusive use of a single group of proteases, matrix metalloproteinases (MMPs), which are known for their broad substrate specificity. Such enzymes are not suitable models for evaluating general mechanisms of substrate selectivity, as their activity does not follow well-defined or highly selective substrate recognition rules. Importantly, no MMPs exhibit the high substrate specificity required to serve as a robust basis for predictive modeling.

Furthermore, the *in silico* results have not been sufficiently or reliably compared and validated against existing experimental data, particularly those obtained using proteomic techniques and tools from chemical biology. The absence of such comprehensive validation significantly reduces the biological and predictive value of the proposed model. To substantiate the proposed approach, it is essential to test the model across a diverse set of protease classes, each representing different substrate-binding modes, hydrolysis mechanisms, and patterns of subsite specificity (for example, cathepsins, caspases, kallikreins and SENPs). A thorough analysis of subsite selectivity across all binding pockets is also necessary for each of these enzymes classes. Only by comparing *in silico* predictions with experimentally established specificity profiles—derived from proteomics and combinatorial chemistry—can the practical utility of the model be properly assessed.

In my opinion, the manuscript in its current form does not meet the publication standards of Nature Communications, as it does not confirm the research hypothesis in a comprehensive manner, nor does it provide sufficient validation of the findings. The authors have limited their analyses to a narrow scope, without contextualizing their results within the broader enzymology of proteases or making adequate use of available comparative data.

In conclusion, while the study may offer a thought-provoking theoretical concept and a potential starting point for future investigations, its current scope and level of validation make it more appropriate for publication in a specialized journal rather than as a full-fledged research article of

broad relevance. Therefore, I do not recommend this manuscript for publication in its current form.

We address these comments in the following point-by-point response, after breaking the review into individual points. We note that many of these comments lack substantive insight or concrete examples of what is missing from the paper and how to improve it.

Comment #1:

“The central research hypothesis—that substrate specificity can be reliably predicted using the proposed model—has not been convincingly validated.”

The central research goal of our work is to design an end-to-end AI pipeline for the design of protease substrates, and to apply that pipeline to matrix metalloproteinases with the hypothesis that it can enhance the scale, tunability, and efficiency of substrate design. Our study shows that CleaveNet can design effective, selective substrates for matrix metalloproteinases (MMPs), with evidence of generalizability across different MMPs. The results are validated by both *in silico* and *in vitro* experiments, as recognized by both Reviewers 1 and 3.

Comment #2:

“A critical limitation of the study is the exclusive use of a single group of proteases, matrix metalloproteinases (MMPs), which are known for their broad substrate specificity. Such enzymes are not suitable models for evaluating general mechanisms of substrate selectivity, as their activity does not follow well-defined or highly selective substrate recognition rules. Importantly, no MMPs exhibit the high substrate specificity required to serve as a robust basis for predictive modeling”

Our work is a focused study of MMPs as an exemplary enzyme class, serving as a proof-of-concept of the CleaveNet approach more generally. The focus on MMPs is because MMPs play critical roles in biology and disease (Cabral-Pacheco et al., 2020, Dudani et al., 2018). While they are known for their broad substrate specificity, the ability to engineer MMP-selective substrates is an active research area, given the biological and therapeutic importance of these enzymes (Kasperkiewicz et al, 2017, Kukreja et al., 2015, Eckhard et al., 2016, Ratnikov et al, 2014).

The goal of our work is to present a tractable AI-driven approach for substrate design, with focus and application to matrix metalloproteinases. This is clearly stated in our **Abstract:**

“Identifying substrates that are efficiently and selectively cleaved by target proteases is essential for studying protease activity and for harnessing their activity in protease-activated diagnostics and therapeutics. However, the vast design space of possible substrates (c.a. 20¹⁰ unique amino acid combinations for a 10-mer peptide) and the limited accessibility of high-throughput activity profiling

tools hinder the speed and success of substrate design. We present CleaveNet, an end-to-end AI pipeline for the design of protease substrates. Applied to matrix metalloproteinases, CleaveNet enhances the scale, tunability, and efficiency of substrate design."

Given the scarcity of publicly available datasets containing broad substrate libraries across multiple enzyme families, we demonstrate proof-of-concept for MMPs, for which a dataset with substantial training data across 18 distinct MMPs was available. Our analyses successfully identify molecular patterns associated with MMP13 efficient (**Figure 5**) and selective (**Figure 7F**) substrates, indicating that our approach serves as a robust basis for the effective design of MMP substrates.

Furthermore, the comment that "[MMPs] are not suitable models for evaluating general mechanisms of substrate selectivity, as their activity does not follow well-defined or highly selective substrate recognition rules" is, in our opinion, not well-founded. There are well-established motifs for MMP cleavage (Overall et al., 2002, Eckhard et al., 2016), even if this class of enzymes is not notable for its selectivity. More generally, we again re-emphasize the goal of our work and the CleaveNet approach: to build and leverage an AI pipeline to learn the cleavage preferences of proteases and use these learned patterns to guide substrate design according to specific attributes, such as selectivity for a target protease relative to other proteases but also efficiency. Our successful design and validation of multiple substrates uniquely cleaved by MMP13 demonstrates that, despite the relative promiscuity of MMPs, it is possible to identify or design substrates that are preferentially cleaved by one MMP over others.

We stress that these considerations, together with our empirical results, demonstrate that MMPs are a robust basis for these design questions and that CleaveNet can enable the successful design of efficient and selective MMP substrates.

Comment #3:

"Furthermore, the in silico results have not been sufficiently or reliably compared and validated against existing experimental data, particularly those obtained using proteomic techniques and tools from chemical biology. The absence of such comprehensive validation significantly reduces the biological and predictive value of the proposed model"

In this study, we assess the predictive performance of the CleaveNet Predictor with two independent, experimentally derived datasets: the mRNA-display test set, and an independent, out-of-distribution fluorescence test set of 71 Fluorescence Resonance Energy Transfer (FRET)-based substrates (**Figure 2**). Both the mRNA-display and fluorescence-derived datasets are sets of existing experimental data. Both mRNA display and FRET probes are tools from chemical

biology, and we use both sets of experimental data to validate our approach, refuting the reviewer's claim that we do not leverage experimental data for validation.

Regarding proteomic techniques, CleaveNet is focused on synthetic peptide substrates for proteases, versus data from proteomics that measure levels of endogenous proteins. This delineation and our rationale for focusing on synthetic substrates is discussed in our manuscript **Introduction** lines 40-43:

"The most common approach for substrate nomination entails surveying the literature for existing substrates, informed by cleavage sites in naturally-occurring proteins (23). However, this search is inefficient and seldom results in synthetic substrates with desired cleavage profiles (19)."

In addition, our end-to-end CleaveNet pipeline yielded substrate designs that were synthesized as FRET-substrates and validated *in vitro* against 12 recombinant MMPs (**Figure 5, Figure 7**). This is a point of prospective experimental validation via a chemical biology approach that supports both the biological and predictive values of the CleaveNet pipeline.

Comment #4:

"To substantiate the proposed approach, it is essential to test the model across a diverse set of protease classes, each representing different substrate-binding modes, hydrolysis mechanisms, and patterns of subsite specificity (for example, cathepsins, caspases, kallikreins and SENPs)"

Our work is a focused study of MMPs as an exemplary enzyme class, serving as a proof-of-concept of the CleaveNet approach more generally; please see reply to comment #2 for our reasoning for this. We believe developing and testing CleaveNet models for other protease classes to be beyond the scope of this work, and suggest this as a line for future work in our manuscript Discussion, given that the CleaveNet approach should be generalizable. This exact point is raised in our **Discussion** lines 485-488:

"Second, CleaveNet is so far only validated for MMP substrate design. Efforts to create similar datasets and models across other protease subclasses could greatly enhance CleaveNet's reach and its utility."

Comment #5:

"A thorough analysis of subsite selectivity across all binding pockets is also necessary for each of these enzymes classes."

We address subsite selectivity across different MMPs in **Figure 4** to demonstrate how MMPs from different functional subclasses demonstrated different cleavage preferences. **Figure 4** is reproduced below, and its results are extensively discussed in the section **Generated sequences recapitulate biologically relevant cleavage patterns across MMP functional classes**.

Figure 4: Generated sequences recapitulate biologically-relevant cleavage patterns across MMP functional classes. (A) IceLogos for individual MMPs, grouped by relevant subclasses, are shown for the top 100 Z-scoring sequences in each set: CleaveNet-generated (green boxes) and mRNA-display training (blue boxes). The IceLogos are normalized by amino acid frequencies from the mRNA-display set. **(B)** Histograms of 3-mer frequencies shared across the top 100 Z-scoring MMP13 sequences in each of the CleaveNet generated (top), mRNA-display training (middle), and site-independent baseline (bottom) sets. The frequencies of the top-5 occurring 3-mers in each set are summarized as tables on the right, with shared 3-mers between the CleaveNet generated and mRNA-display sets **bolded**. **(C)** Heat map, colored by CleaveNet-predicted cleavage score and annotated by hierarchical clustering, of the 25-top Z-scoring substrates per MMP, including the similarity cutoff and subsequent clustering of proteases into 5 groups with shared phylogeny.

Comment #6:

Only by comparing *in silico* predictions with experimentally established specificity profiles—derived from proteomics and combinatorial chemistry—can the practical utility of the model be properly assessed

We provide a direct comparison between *in silico* predictions and their true experimental cleavage profiles as presented in **Figure 2**, **Table S1**, and **Figures S2–12**. We stress that mRNA-display is a combinatorial technique.

Further, we validate CleaveNet-generated designs *in vitro* (**Figure 5** and **Figure 7**). The performance of CleaveNet is evaluated against the top-performing substrates identified within the training dataset, which were obtained using an mRNA display assay employing a library constructed from random combinations of accepted residues at each individual P5'-P5 position (Kukreja et al., 2015). In addition, we benchmark the model against sequences obtained through site-independent random sampling of amino acid distributions from highly efficient and selective substrates.

The extensive design space (c.a. 20^{10}) of potential MMP-cleavable 10-mer substrates inherently limits the number of candidates that can be experimentally screened with fluorogenic assays that confidently allow assessment of true cleavage. Our study demonstrates that CleaveNet can help explore these complex chemical landscapes by learning from sparse datasets, eliminating the need to explicitly explore the entire ten-trillion-parameter design space with experiments.

Reviewer #3

The authors report the development of sequence-based AI models for predicting and designing protease specificity, focusing on MMPs. The studies are designed creatively, performed carefully, interpreted thoughtfully, and reported succinctly.

We thank the reviewer for their constructive feedback and positive comments on our study.

Comment #1:

For in silico evaluation of protein-related problems, there is always the question of data leakage leading to memorization effects. The experimental results showing efficacy mitigate this concern to a great extent, but instead of using Logo plots, the authors could perform additional analysis of how much training and test sets overlap and implications of that, as well as comment on dataset balancing?

We appreciate the reviewer's comment, which was shared by Reviewer #1. To mitigate concerns over data leakage, we prepared a new homology-filtered mRNA-display test set and re-analyzed all the results of the paper using this new test set. Briefly, we computed the minimum Levenshtein distance between each test sequence and the training set, to find the closest train match to each test sequence. We find that of the 3,717 original test sequences, 816 sequences have some shared homology (distance < 3) to sequences in the train set. We removed these sequences from the test set, leaving 2,901 non-overlapping sequences to use as a new homology-filtered mRNA test set.

Using this new test set we re-computed all analyses in the paper and found that overall, there are minimal to negligible differences in the results when using this new homology-filtered test set (**Table R2, R3, R4** in the response to Reviewer #1).

To mitigate any possible concerns surrounding the quality of the test set, we include these new analyses in the paper, using the new homology-filtered mRNA-display test set in place of the old random test set. This includes updates to **Figure 2B, 2C, 2D; Table S1; Figure S2, S3, S4** and **S5**. We refer this reviewer to the response to Reviewer #1 where we replicate these results in the rebuttal. We believe these new results support the robustness of CleaveNet's predictive performance and also address any concerns about potential data leakage or memorization.

Comment #2:

It is great to see that the generated sequences have higher catalytic efficiency. Can the authors identify why this is the case? What is the expectation from chance?

We thank the reviewer for this observation, which also came to our surprise initially. Indeed, CleaveNet-generated efficient sequences achieve significantly higher efficiency than the mRNA-display efficient controls (**Figure 5B**), and a subset of CleaveNet-generated selective sequences (**Figure 7D**; upper right quadrant) achieve superior efficiency, as well. We hypothesize that this is because CleaveNet models effectively learn sequence-level patterns of MMP cleavage, driven by non-linear interactions across multiple residues, and their associations with different cleavage profiles. This enables use of CleaveNet to oversample sparse sequence spaces of interest (e.g. high MMP13 efficiency or selectivity for which 20,000 sequences each were generated from sets of 50 seeds), enabling discovery of novel sequences that achieved levels of performance unseen in the training set.

Expectation from chance is evaluated with side-independent baselines in **Figures 5 and 7**; these site-independent baselines involve sampling independently from the position-wise amino acid distributions and exhibit very poor performance. Note that these are stronger baselines than uniform random sampling or random sampling from the aggregate amino acid distribution (aggregated across positions) from the mRNA-display dataset. As discussed in the main text, this poor performance is attributed to these site-independent models failing to capture critical site-cooperativity patterns learned by CleaveNet. We refer the reviewer to our **Results** section **Generated sequences recapitulate biologically relevant cleavage patterns across MMP functional classes** for further discussion on how our models capture site-cooperativity effects.

To shed further light on this observation raised by the reviewer, we have added the following text to the Discussion on lines 456-458:

“These findings highlight the potential utility of CleaveNet for oversampling sparse sequence spaces of interest, to identify new and even superior substrates with novel motifs and desirable cleavage profiles.”

Comment #3:

Have the authors considered using structural models to gain insights into successful and failed predictions?

We thank the reviewer for this suggestion. In response, we used AlphaFold3 (Abramson et al. *Nature* 2024) to model the catalytic domain of human MMP13 with 2 Zn²⁺ ions and modeled 9 selective substrates (DL43, DL29, DL38, DL60, DL61, DL33, DL89, DL41, DL28); 9 efficient substrates (DL73, DL6, DL50, DL49, DL52, DL3, DL57, DL14, DL20); 5 efficient & selective substrates (DL5, DL16, DL45, DL42, DL48) and 3 negative control substrates (DL63, DL64, DL65). We then analyzed

protein–substrate contacts using the Protein–Ligand Interaction Profiler (Schake, Bolz et al. *NAR* 2025).

From this analysis, we made a few noteworthy observations:

- Pi-stacking interactions between the substrate and catalytic histidine H119, which coordinates the active-site zinc ion, were less common in efficient substrates than in selective substrates.
- Salt bridges involving catalytic histidines (H119, H123, H129), which coordinate the active-site zinc, were less frequent in efficient substrates compared to selective ones.
- A hydrophobic interaction between L81 (L185 in full length hMMP13), which is described as part of the S3' pocket of MMP13 (Stura, Bellamy et al., 2013), and positions 5 or 7 of the substrate was absent in the efficient & selective group, but present in many of the substrates that were efficient *or* selective.

The number of substrates in each group, especially the “efficient & selective” group, is small; therefore, we feel these observations should be interpreted as preliminary and thus keep them in our rebuttal versus also incorporating them into the manuscript. With this caveat, these observations may suggest a possible mechanistic explanation for determinants of efficiency and selectivity: interactions that stabilize zinc coordination (pi-stacking, salt bridges) appear to bias toward selectivity, while their absence favors efficiency. The specifics of the L81–substrate hydrophobic interaction may be important for generation of substrates that are both efficient and selective. Broadly, we are excited by the potential for these types of analyses and structural modeling to generate testable hypotheses around the molecular determinants of efficiency versus selectivity, a direction we hope to explore in future studies.

We expand on this in our manuscript Discussion, lines 491-496, with new additions in **bold**:

*“With respect to MMP13 substrate design, **structure-based** docking analyses could reveal **greater** mechanistic understanding of **the molecular determinants of MMP13 substrate recognition**, and the selectivity and efficiency of designed substrates could continue to be refined by **such structural hypotheses**, additional rounds of conditioning around top hits, or **incorporation of non-natural amino acids.**”*

Comment #4:

Also please cite the following protease specificity measurement and prediction works:

<https://pubmed.ncbi.nlm.nih.gov/23589865/>

<https://pubmed.ncbi.nlm.nih.gov/27932294/>

<https://pubmed.ncbi.nlm.nih.gov/30587591/>

<https://pubmed.ncbi.nlm.nih.gov/37729196/>

<https://pubs.acs.org/doi/10.1021/acssynbio.0c00452>

We thank the reviewer for pointing out these citations. After review of these works, we recognize the relevance of Pethe et al. 2017, which uses a structure and energetics-based approach for predicting protease substrate specificity, and Pethe et al. 2019, which uses structure-based models and supervised learning to predict the cleavability of candidate substrates for the HCV protease, to CleaveNet and agree that they should be referenced in the manuscript. We have referenced them in the manuscript Introduction as follows on lines 52-55:

*"Methods leveraging inferred substitution matrices (29–31), **structural and energetics patterns (32, 33)**, or supervised learning (34–38) have focused on identifying substrate cleavage sites to predict whether a substrate will be recognized by a target protease."*

The remaining works (Yi et al. PNAS 2013, Lu et al. PNAS 2023, Denard et al. ACS Synthetic Biology 2021) are protease-centric, with a specific focus on protease engineering. In particular, the works of Yi et al. and Denard et al. involve engineering improved protease variants via synthetic biology approaches, with applications to TEV protease engineering. Given that CleaveNet is substrate-centric and focused on substrate design, not protease design, and furthermore does not consider the TEV protease or TEV substrates, these works (Yi et al. PNAS 2013, Lu et al. PNAS 2023, Denard et al. ACS Synthetic Biology 2021) are not directly relevant to CleaveNet, and therefore we do not reference them in our manuscript.

Reviewer #4

References

1. Abramson, J., et al. "Accurate structure prediction of biomolecular interactions with AlphaFold 3." *Nature* 630.8016 (2024): 493-500.
2. Cabral-Pacheco, G.A., et al. "The Roles of Matrix Metalloproteinases and Their Inhibitors in Human Diseases." *Int J Mol Sci.* (2020): 2020;21(24):9739.
3. Denard, Carl A., et al. "YESS 2.0, a tunable platform for enzyme evolution, yields highly active TEV protease variants." *ACS Synthetic Biology* 10.1 (2021): 63-71.
4. Dudani, J.S., et al. "Harnessing Protease Activity to Improve Cancer Care" *Annual Review of Cancer Biology* 2 (2018): 353-376.
5. Eckhard, U. et al. "Active site specificity profiling datasets of matrix metalloproteinases (MMPs) 1, 2, 3, 7, 8, 9, 12, 13 and 14." *Data in Brief* 7 (2016): 299-310.
6. Kasperkiewicz, P. et al. "Emerging challenges in the design of selective substrates, inhibitors and activity-based probes for indistinguishable proteases." *The FEBS Journal* 284.10 (2017): 1518-1539.
7. Kukreja, M. et al. "High-throughput multiplexed peptide-centric profiling illustrates both substrate cleavage redundancy and specificity in the MMP family." *Chemistry & Biology* 22.8 (2015): 1122-1133.
8. Li, F. et al. "ProsperousPlus: a one-stop and comprehensive platform for accurate protease-specific substrate cleavage prediction and machine-learning model construction." *Briefings in Bioinformatics* 24.6 (2023): bbad372.
9. Lu, Changpeng, et al. "Prediction and design of protease enzyme specificity using a structure-aware graph convolutional network." *Proceedings of the National Academy of Sciences* 120.39 (2023): e2303590120.
10. Pethe, M. et al. "Large-scale structure-based prediction and identification of novel protease substrates using computational protein design." *Journal of Molecular Biology* 429.2 (2017): 220-236.
11. Pethe, M. et al. "Data-driven supervised learning of a viral protease specificity landscape from deep sequencing and molecular simulations." *Proceedings of the National Academy of Sciences* 116.1 (2019): 168-176.

12. Ratnikov, B.I. et al. "Basis for substrate recognition and distinction by matrix metalloproteinases." *Proceedings of the National Academy of Sciences* 111.40 (2014): E4148-E4155.
13. Schake, P., et al. "PLIP 2025: introducing protein–protein interactions to the protein–ligand interaction profiler." *Nucleic Acids Research* (2025): gkaf361.
14. Stura, Enrico A., et al. "Crystal structure of full-length human collagenase 3 (MMP-13) with peptides in the active site defines exosites in the catalytic domain." *The FASEB Journal* 27.11 (2013): 4395.
15. Yi, Li, et al. "Engineering of TEV protease variants by yeast ER sequestration screening (YESS) of combinatorial libraries." *Proceedings of the National Academy of Sciences* 110.18 (2013): 7229-7234.